# Mutational analysis to explore long-range allosteric couplings involved in a pentameric channel receptor pre-activation and activation

Solène N Lefebvre[1,2], Antoine Taly[3,4]*, Anaïs Menny[1,2†], Karima Medjebeur[1], Pierre-Jean Corringer[1]*

[1]Institut Pasteur, Université de Paris, CNRS UMR 3571,Channel-Receptors Unit, Paris, France; [2]Sorbonne Université, Collège doctoral, Paris, France; [3]Institut de Biologie Physico-chimique, Fondation Edmond de Rothschild, PSL Research University, Paris, France; [4]Laboratoire de Biochimie Théorique, CNRS, Université de Paris, UPR 9080, Paris, France

*For correspondence:
antoine.taly@ibpc.fr (AT);
pjcorrin@pasteur.fr (P-JC)

Present address: †Institut de Génomique Fonctionnelle, Université de Montpellier, CNRS, INSERM, Montpellier, France

Competing interest: The authors declare that no competing interests exist.

**Abstract** Pentameric ligand-gated ion channels (pLGICs) mediate chemical signaling through a succession of allosteric transitions that are yet not completely understood as intermediate states remain poorly characterized by structural approaches. In a previous study on the prototypic bacterial proton-gated channel GLIC, we generated several fluorescent sensors of the protein conformation that report a fast transition to a pre-active state, which precedes the slower process of activation with pore opening. Here, we explored the phenotype of a series of allosteric mutations, using simultaneous steady-state fluorescence and electrophysiological measurements over a broad pH range. Our data, fitted to a three-state Monod-Wyman-Changeux model, show that mutations at the subunit interface in the extracellular domain (ECD) principally alter pre-activation, while mutations in the lower ECD and in the transmembrane domain principally alter activation. We also show that propofol alters both transitions. Data are discussed in the framework of transition pathways generated by normal mode analysis (iModFit). It further supports that pre-activation involves major quaternary compaction of the ECD, and suggests that activation involves principally a reorganization of a 'central gating region' involving a contraction of the ECD β-sandwich and the tilt of the channel lining M2 helix.

## Introduction

Pentameric ligand-gated ion channels (pLGICs) mediate fast synaptic communication in the brain. In mammals, this family includes the excitatory nicotinic acetylcholine (ACh) and serotonin receptors (nAChRs and 5-HT$_3$Rs) as well as the inhibitory γ-aminobutyric acid (GABA) and glycine receptors (GABA$_A$Rs and GlyRs) (*Jaiteh et al., 2016*). pLGICs are also present in bacteria, notably with the pH-gated channels GLIC (*Bocquet et al., 2007*) and sTeLIC (*Hu et al., 2018*), the GABA-gated channel ELIC (*Zimmermann and Dutzler, 2011*), and the calcium-modulated DeCLIC (*Hu et al., 2020*).

pLGICs physiological function is mediated by alternating between different allosteric conformations in response to neurotransmitter binding. Initially, a minimal four-state model could describe the main allosteric properties of the muscle-type nAChR (*Heidmann and Changeux, 1980*; *Sakmann et al., 1980*). In this model, the ability of ACh binding to activate the nAChR involves a resting- to active-state transition, and prolonged ACh occupancy promotes a biphasic desensitization process. Subsequently, kinetic analysis of the close-to-open transitions recorded by single-channel electrophysiology

revealed multiple additional states that are required to account for the observed kinetic patterns. For activation, short-lived intermediate 'pre-active' states named 'flipped' (*Lape et al., 2008*) and 'primed' (*Mukhtasimova et al., 2009*) were included in the kinetic schemes of the GlyRs and nAChRs, while rate-equilibrium free-energy relationship analysis of numerous mutants of the nAChR suggested passage through four brief intermediate states (*Gupta et al., 2017*). Likewise, analysis of single-channel shut intervals during desensitization is described by the sum of four or five exponential components, suggesting again additional intermediate states (*Elenes and Auerbach, 2002*). Kinetics data thus show that pLGICs go through complex structural reorganizations during both activation and desensitization. These events are at the heart of the protein's function, allowing coupling between the neurotransmitter site and the ion channel gate which are separated by a distance of more than 50 Å.

The past decade has seen great structural biology efforts to increase our understanding of the molecular mechanisms involved in gating (*Nemecz et al., 2016*). At least one structure of each major member of prokaryotic (*Hilf and Dutzler, 2008*; *Bocquet et al., 2009*; *Hu et al., 2018*; *Hu et al., 2020*) and eukaryotic pLGICs (*Althoff et al., 2014*; *Du et al., 2015*; *Polovinkin et al., 2018*; *Gharpure et al., 2019*; *Masiulis et al., 2019*) have been resolved by X-ray crystallography or cryo-electron microscopy (cryoEM). They highlight a highly conserved 3D architecture within the family. Each subunit contains a large extracellular domain (ECD) folded in a β-sandwich and a transmembrane domain (TMD) containing four α-helices, with the second M2-helix lining the pore. However, the physiological relevance of these structures or their assignment to particular intermediates or end-states in putative gating pathways remains ambiguous and poorly studied. Conversely, it is possible that intermediate conformations, unfavored by crystal packing lattice or under-represented in receptor populations on cryoEM grids, are missing in the current structural galleries.

Understanding the allosteric transitions underlying gating thus requires complementary techniques, where the protein conformation can be followed in near-physiological conditions, that is at non-cryogenic temperature on freely moving protein, and over a broad range of ligand concentrations. To this aim, we previously developed the tryptophan/tyrosine induced quenching technique (TrIQ) on GLIC (*Menny et al., 2017*), a proton-gated channel (*Parikh et al., 2011*; *Bocquet et al., 2007*; *Laha et al., 2013*; *Gonzalez-Gutierrez et al., 2017*). In this technique, the protein is labeled with a small fluorophore(bimane, and collisional quenching by a neighboring indole (tryptophan) or phenol (tyrosine) moiety is used to report on changes in distance between two residues within the protein over a short distance range of 5–15 Å (*Mansoor et al., 2002*; *Mansoor et al., 2010*; *Jones Brunette and Farrens, 2014*). Bimane-quencher pairs on GLIC combined with kinetic analysis allowed us to characterize pre-activation motions occurring early in the conformational pathway of activation (*Figure 1A*). We found that they occur at lower proton concentrations than pore opening, and are complete in less than a millisecond, much faster than the rise time of the active population that occurs in the 30–150 ms range in electrophysiology recordings (*Laha et al., 2013*).

Here, to explore the conformational landscape of GLIC during pH-gating, we further exploited the TrIQ approach. We performed electrophysiological and fluorescence quenching experiments on a series of allosteric mutants of GLIC, as well as in the presence of the general anesthetic propofol. We modeled the whole data set with a three-state allosteric model comprising a resting state, a pre-active state, and an active state. To help the interpretation of the fluorescence quenching data into structural terms, we built atomistic models of the various bimane-labeled proteins, and computed their gating transition pathways using iMODfit. Our results indicate that mutations alter the function via distinct mechanisms and differentially displace the allosteric equilibria involved in fluorescence quenching and electrophysiology recordings. This supports that pre-activation involves a major quaternary compaction of the ECD, and suggests that activation involves principally a reorganization of a 'central gating region' involving a contraction of the ECD β-sandwich and the tilt of the channel lining M2 helix.

## Results

### Fluorescence and electrophysiological measurements

#### Quenching pairs used in the study

In our previous fluorescence quenching experiments, a bimane fluorophore was introduced on GLIC by covalent labeling on an engineered cysteine, after mutation of the single endogenous cysteine C27S. A Trp or Tyr quenching residue was incorporated when necessary to generate a quenching pair.

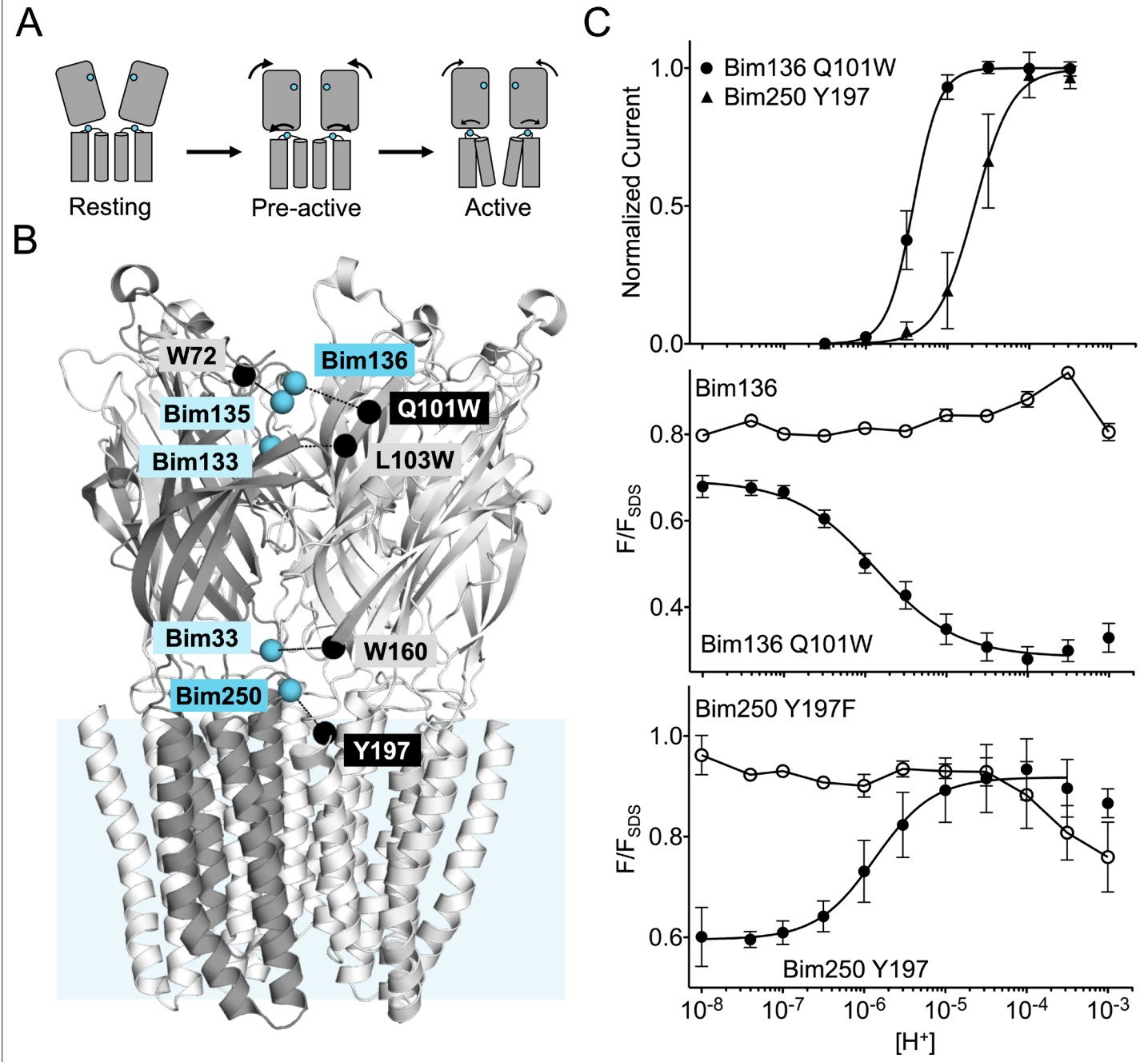

**Figure 1.** Electrophysiological and fluorescence characterization of the quenching pairs of GLIC. (**A**) Scheme for GLIC activation, showing first a pre-activation step involving full compaction of the ECD and motion of the M2-M3 loop as monitored by fluorescence, followed by a pore opening step. Blue spheres indicate the location of sensors Bim136-Q101W and Bim250-Y197 used thereafter in this study. (**B**) GLIC-pH 4 (pdb code 4HFI) structure side view, the light blue rectangle represents the position of the membrane. Quenching pairs generated in our previous study (*Menny et al., 2017*) are highlighted: blue spheres show the $C_\alpha$ of the residues that were mutated into cysteines and bimane labeled (Bim33, Bim133, Bim135, Bim136, and Bim250), black spheres show the $C_\alpha$ of the quenchers (W160, L103W, W72, Q101W, and Y197). (**C**) pH-dependent response curves of Bim136-Q101W and Bim250-Y197 sensors, by electrophysiology after labeling (top panel) and with bimane fluorescence quenching (lower two panels). Fluorescence data are shown normalized to the fluorescence of the denatured protein ($F_{SDS}$), bimane fluorescence is shown without quencher (O) and in presence of the quencher (●). ECD, extracellular domain.

The online version of this article includes the following figure supplement(s) for figure 1:

**Source data 1.** Fluorescence quenching and electrophysiological current measurements of the different mutants tested.

**Figure supplement 1.** Fluorescence quenching data from mutants with the Bim135 W72 sensor.

**Figure supplement 2.** pH-dependent curves comparison.

We created five quenching pairs (*Figure 1B*): three are located across the ECD interface and report on a quaternary compaction following pH drop (Bim136-Q101W, Bim133-L103W, and Bim33-W160), one reports on a tertiary reorganization at the top of the ECD (Bim135-W72), and one reports on the outward movement of the M2-M3 loop at the ECD-TMD interface (Bim250-Y197). In the present study, we used the Bim136-Q101W as sensor of the ECD compaction, along with Bim250-Y197 as a sensor of the M2-M3 loop motion. We also investigated in detail Bim135-W72, but the complex results for this pair precluded clear conclusions. The related data are thus presented and discussed in *Figure 1—figure supplement 1*.

To accurately compare mutants, we first measured detailed pH-dependent fluorescence and electrophysiological curves (*Figure 1C*). Fluorescence was measured in steady-state conditions on detergent (DDM)-purified protein, and normalized to the fluorescence intensity under denaturing conditions (1 % SDS), as previously described (*Menny et al., 2017*). GLIC allosteric transitions are particularly robust in different lipid/detergent conditions (*Sauguet et al., 2014*; *Carswell et al., 2015*) and DDM-purified protein yielded similar results to that of azolectin-reconstituted protein (*Menny et al., 2017*), while allowing better reproducibility. For both sensors, we confirmed that the pH-dependent fluorescence changes are essentially abolished when mutating the quenching partner to phenylalanine, which does not quench bimane fluorescence (*Mansoor et al., 2002*). We also confirmed that pH-dependent quenching curves for Bim136-Q101W and Bim250-Y197 display higher sensitivity (especially for Bim250-Y197) and lower apparent cooperativity than the pH-dependent activation curves recorded by electrophysiology (*Figure 1—figure supplement 2*).

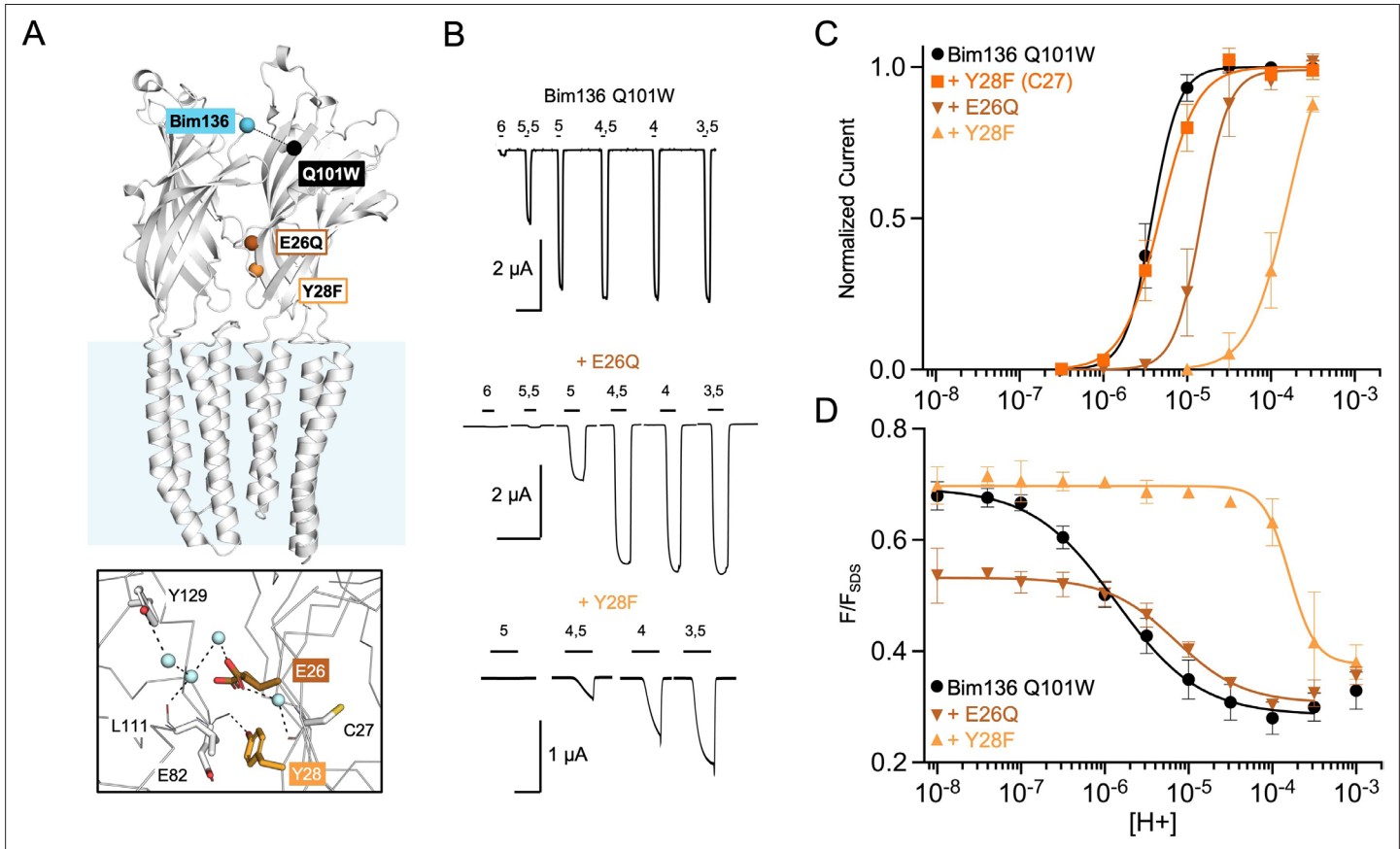

**Figure 2.** Allosteric coupling within the ECD. (**A**) Structure of two monomers of GLIC pH 4 (4HFI) showing positions of the fluorescence sensor (Bim136-Q101W) and the two mutated residues at the bottom of the ECD resulting in a partial loss of function. The lower panel shows a zoom on the interface with E26 and Y28 residues and their interactions with surrounding residues and a network of water molecules (blue spheres). (**B**) Electrophysiological recordings in oocytes of the mutants labeled with bimane showing shifted responses to higher proton concentrations in comparison with GLIC Bim136-Q101W. pH applications are shown above each trace and the horizontal scale represents 1 min of recording. Graphs represent pH-dependent curves showing a shift to higher proton concentrations in electrophysiological responses (**C**) and fluorescence quenching responses (**D**) for both mutants. ECD, extracellular domain.

## The quaternary compaction at the ECD top is strongly allosterically coupled with the lower part of the ECD interface

Using the Bim136-Q101W conformational sensor, we first investigated allosteric mutants located at the inter-subunit interface in the lower part of the ECD (*Figure 2A*). We previously showed that E26Q produces a decrease in $pH_{50}$ for activation (*Nemecz et al., 2017*), a phenotype that is conserved here on the Bim136-Q101W background (*Figure 2B and C*). The fluorescence quenching curve of Bim136-Q101W-E26Q also shows a decrease in $pH_{50}$ (*Figure 2D*), and the $\Delta pH_{50}$ between Bim136-Q101W and Bim136-Q101W-E26Q are nearly identical in electrophysiology and fluorescence (−0.59 and −0.57, respectively; *Table 1*). Interestingly, the Bim136-Q101W-E26Q fluorescence quenching curve has a remarkable feature as compared to most mutants investigated thereafter: in the pH 7–8 range, where the pH-dependent fluorescence quenching is not yet observed, the fluorescence ($F/F_{SDS}$) is significantly lower ($F_0$=0.53) as compared to the Bim136-Q101W alone ($F_0$=0.71). This suggests that substantial quenching is present at neutral pH and that E26Q not only alters the allosteric transition, but also modifies the conformation of the resting state itself which appears to be more compact when the E26Q mutation is present.

Another mutation, Y28F, was reported to produce a moderate gain of function on a wild-type background (*Nemecz et al., 2017*). Surprisingly, mutating Y28F in the Bim136-Q101W (C27S) background yields a drastic loss of function characterized by a slow activating receptor and a marked decrease in $pH_{50}$ (*Figure 2B and C*; *Table 1*). However, mutating back the C27 endogenous cysteine (Bim136-Q101W-Y28F (C27)) reverses the phenotype to that of Bim136-Q101W (C27S) (*Figure 2C* and *Table 1*), demonstrating that this loss of function is due to the combination of the C27S and Y28F mutations. In fluorescence, the quenching curve of Bim136-Q101W-Y28F (C27S) also shows a large decrease in $pH_{50}$, associated with an apparent higher cooperativity. Again, the $\Delta pH_{50}$ is in the same range in fluorescence quenching (–2.2) and in electrophysiology (more than −1.5, the plateau could not be reached with this mutant preventing accurate measurement of the $pH_{50}$).

In conclusion, the quaternary compaction of the top of the ECD, monitored with the Bim136-Q101W sensor, is strongly coupled in an allosteric manner with the lower part of the ECD interface.

## Long-range allosteric coupling between the TMD and the top of the ECD

To investigate whether allosteric coupling occurs with more distant regions of the protein, we selected three loss of function mutations further away from the Bim136-Q101W pair: D32E near the ECD-TMD interface; H235Q in the middle of the TMD and E222Q, at the bottom of the TMD and lining the pore (*Figure 3A*; *Sauguet et al., 2014*; *Nemecz et al., 2017*).

Performing these mutations on the Bim136-Q101W-C27S background shows overall conservation of their previously published phenotype, with a 0.7 unit (D32E and E222Q) and 1.3 unit (H235Q) decreases in the $pH_{50}$ of activation as compared to Bim136-Q101W-C27S (*Figure 3B and C*; *Table 1*). The fluorescence quenching curves are also shifted to lower $pH_{50}$s, with $\Delta pH_{50}$s of 0.3–0.5 (D32E and E222Q) and 0.85 (H235Q) (*Figure 3D*).

The quenching data thus reveal an allosteric coupling between both ends of the protein, since the structural perturbations performed around the TMD are transmitted to the top of the ECD, impairing its compaction. However, as opposed to the ECD mutations E26Q and Y28F/C27S, these mutations have a stronger effect on the $pH_{50}$ of the electrophysiological response as compared to fluorescence quenching. It thus suggests that both processes are not fully coupled for mutations further away from the sensor site.

## Total loss of function mutations differentially alter ECD and TMD allosteric motions

To further explore the allosteric coupling within GLIC, we extended the analysis to mutations known to strongly or completely prohibit channel opening (*Figure 4A and B*). We selected three mutants: H235F, L157A, and L246A which show robust surface expression and no substantial current in oocytes (*Figure 4C* and *Figure 4—figure supplement 1*). For those mutants, in addition to electrophysiological recordings and fluorescence quenching measurements on Bim136-Q101W and Bim135-W72 (*Figure 4D* and *Figure 1—figure supplement 1*), we also monitored the motion of the M2-M3 loop with Bim250-Y197 (*Figure 4E*).

**Table 1.** pH-dependence of electrophysiological and fluorescence quenching responses.

$pH_{50}$ and Hill coefficient $n_H$ average and standard deviation values are shown after individual fitting of each measurement. n corresponds to the number of oocytes for electrophysiology and the number of fluorescence measurements, each measurement including values for a full pH range. $F_0$ corresponds to the initial fluorescence value at pH 7/8 and $\Delta F_{max}$ is the maximum variation in fluorescence amplitude within the pH range (absolute values). To reasonably fit Bim136-Q101W + propofol current and Bim136-Q101W-Y28F fluorescence, Hill coefficients have been constrained to 2.5 and below 3 respectively. $\Delta pH_{50}$s are calculated between mutants and their parent construct Bim136-Q101W or Bim250-Y197 (labeled Ref). Their significance was calculated with a one-way ANOVA test using a Dunnett's multiple comparisons test. The p-value is significantly different with p-value≤0.0001 (****), ≤ 0.001 (***), ≤0.01(**), ≤0.05 (*) or not significantly different when p-value>0.05 (ns). NF stands for non-functional and ND for not determined. To compare electrophysiological $pH_{50}$ and fluorescence $pH_{50}$ for each mutant (right column), unpaired t-tests were done with two-tailed p-value and 95 % confidence intervals.

| Mutant | Electrophysiological response bimane labeled | | | | | Fluorescence quenching response in detergent solution | | | | | | | Fluorescence/ electrophysiology | |
|---|---|---|---|---|---|---|---|---|---|---|---|---|---|---|
| | $pH_{50}$ | $n_H$ | n | $\Delta pH_{50}$ | | $pH_{50}$ | $F_0$ | $\Delta F_{MAX}$ | $n_H$ | n | $\Delta pH_{50}$ | | $\Delta pH_{50}$ | |
| Bim136-Q101W C27S | 5.42±0.08 | 2.68±0.33 | 10 | Ref | | 5.85±0.21 | 0.71±0.03 | 0.45±0.06 | 0.77±0.18 | 17 | Ref | – | 0.43 | *** |
| + E26Q | 4.83±0.12 | 2.98±0.64 | 6 | -0.59 | **** | 5.28±0.34 | 0.53±0.02 | 0.22±0.02 | 1.13±0.27 | 4 | -0.57 | ** | 0.45 | * |
| + Y28 F | 3.88±0.08 | 2.63±0.68 | 3 | -1.54 | **** | 3.68±0.34 | 0.70±0.01 | 0.38±0.09 | <3 | 3 | -2.17 | **** | -0.2 | *** |
| + Y28 F & C27 | 5.34±0.11 | 2.03±0.12 | 6 | -0.08 | ns | ND | ND | ND | ND | – | – | – | – | – |
| + D32E | 4.65±0.12 | 2.62±1.51 | 6 | -0.77 | **** | 5.52±0.05 | 0.68±0.02 | 0.38±0.01 | 0.75±0.05 | 3 | -0.33 | ns | 0.87 | *** |
| + E222Q | 4.68±0.09 | 2.51±0.33 | 5 | -0.74 | **** | 5.36±0.14 | 0.66±0.03 | 0.40±0.04 | 0.74±0.01 | 3 | -0.49 | * | 0.68 | *** |
| + H235Q | 4.04±0.21 | 1.19±0.31 | 6 | -1.38 | **** | 5.00±0.09 | 0.70±0.01 | 0.42±0.01 | 0.78±0.01 | 3 | -0.85 | **** | 0.96 | *** |
| + Propofol | 5.16±0.13 | =2.5 | 3 | -0.26 | * | 5.33±0.06 | 0.67±0.02 | 0.38±0.01 | 1.21±0.21 | 4 | -0.52 | ** | 0.17 | ns |
| + H235 Q & propofol | 4.71±0.15 | 1.53±0.48 | 6 | -0.71 | **** | 5.67±0.14 | 0.70±0.01 | 0.43±0.01 | 1.25±0.04 | 3 | -0.18 | ns | 0.96 | *** |
| + H235 F | NF | NF | 3 | – | | 5.25±0.08 | 0.70±0.05 | 0.38±0.04 | 1.06±0.17 | 3 | -0.60 | ** | – | – |
| + L157 A | NF | NF | 3 | – | | 5.42±0.65 | 0.67±0.01 | 0.19±0.09 | 0.71±0.46 | 4 | -0.43 | * | – | – |
| + L246 A | NF | NF | 3 | – | | 4.87±0.14 | 0.65±0.01 | 0.24±0.01 | 0.79±0.12 | 3 | -0.98 | **** | – | – |
| Bim250-Y197 | 4.66±0.18 | 2.20±0.54 | 12 | -0.76 | **** | 5.83±0.17 | 0.59±0.04 | 0.33±0.09 | 1.19±0.28 | 8 | -0.02 | ns | Ref | |
| + H235 F | NF | NF | 3 | – | | 5.40±0.13 | 0.54±0.01 | 0.20±0.01 | 1.16± 0.15 | 4 | – | | -0.43 | ** |
| + L157 A | NF | NF | 3 | – | | 5.81±0.19 | 0.49±0.06 | 0.45±0.07 | 0.64±0.06 | 3 | – | | -0.02 | ns |
| + L246 A | NF | NF | 3 | – | | 5.53±0.03 | 0.64±0.01 | 0.30±0.01 | 1.69±0.29 | 3 | – | | -0.3 | * |

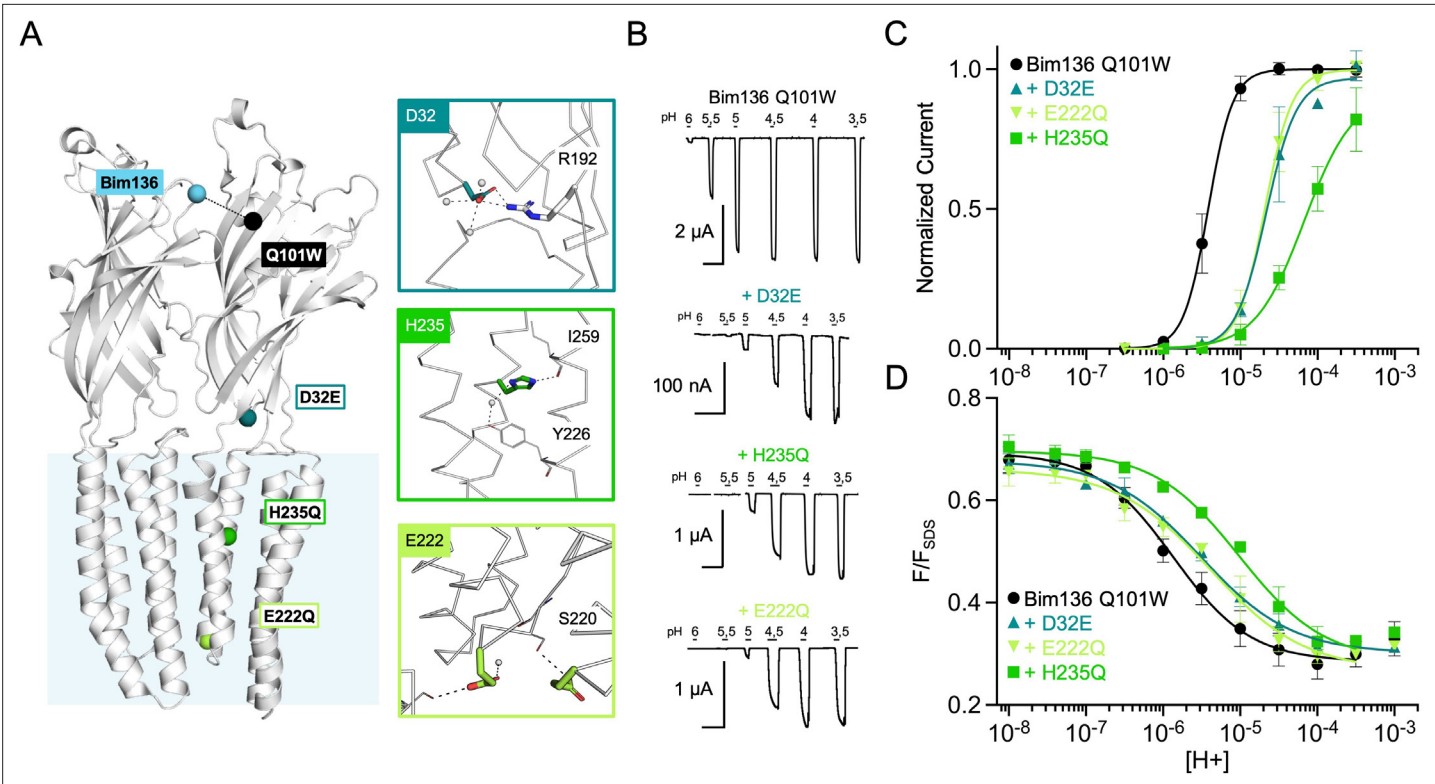

**Figure 3.** Allosteric coupling between the top of the ECD and the TMD. (**A**) Structure of two monomers of GLIC pH 4 (4HFI) showing positions of the fluorescence sensor (Bim136-Q101W) at the top of the ECD and three mutations distributed along the protein. Right panels show zooms on important interactions with the mutated residues. (**B**) Electrophysiological recordings of the three mutants in oocytes, labeled with bimane. Recording of GLIC Bim136-Q101W is shown for comparison. pH applications are shown above each trace and the horizontal scale represents 1 min of recording. pH-dependent curves for electrophysiological response (**C**) and fluorescence quenching (**D**) for the three mutants in comparison with Bim136-Q101W showing a shift to higher proton concentrations of the response for all three mutants. ECD, extracellular domain; TMD, transmembrane domain.

Mutants L157A and L246A reveal unique quenching phenotypes. Combined with Bim136-Q101W, they both show a pH-dependent quenching of fluorescence with a decreased amplitude ($\Delta F_{max}$) associated with a significant decrease in $pH_{50}$ as compared to the Bim136-Q101W background (**Table 1**). In contrast, they only weakly alter the motions at Bim250, which occur with a complete amplitude and small changes in $pH_{50}$. The mutation H235F leads to a phenotype opposite to that of L157A or L246A. Its Bim136-Q101W pH-dependent curve shows a nearly full quenching amplitude together with a decrease in $pH_{50}$, while it impairs the motion of the M2-M3 loop, with only a partial pH-dependent de-quenching at Bim250-Y197.

Thus, while those mutants do not have a measurable access to the active state, they still show allosteric motions as revealed by fluorescence. Unlike the moderate loss of function mutants investigated above, these mutations alter the amplitude of the fluorescence curves, revealing profound changes of either the protein conformations and/or allosteric equilibria.

## Long-range allosteric coupling between the ECD top and propofol

We further used the TrIQ technique to study the mechanism of action of the general anesthetic propofol, an allosteric modulator of GLIC. Propofol binds to at least three main sites within the TMD: one site in the pore itself near the middle of the TMD, and two sites in the upper part of the TMD at intra- or inter-subunit locations (**Figure 5A**). Propofol is an inhibitor of GLIC, but it has been shown to be a potentiator of the H235Q mutant (**Fourati et al., 2018**). We verified that these effects are conserved in the Bim136-Q101W background, with propofol decreasing the $pH_{50}$ of activation of Bim136-Q101W while increasing the $pH_{50}$ of Bim136-Q101W-H235Q (**Figure 5B and C**; **Table 1**). Fluorescence quenching experiments essentially parallel the electrophysiological data. Addition of 100 µM propofol on Bim136-Q101W decreases the fluorescence $pH_{50}$ by half a unit, while it increases

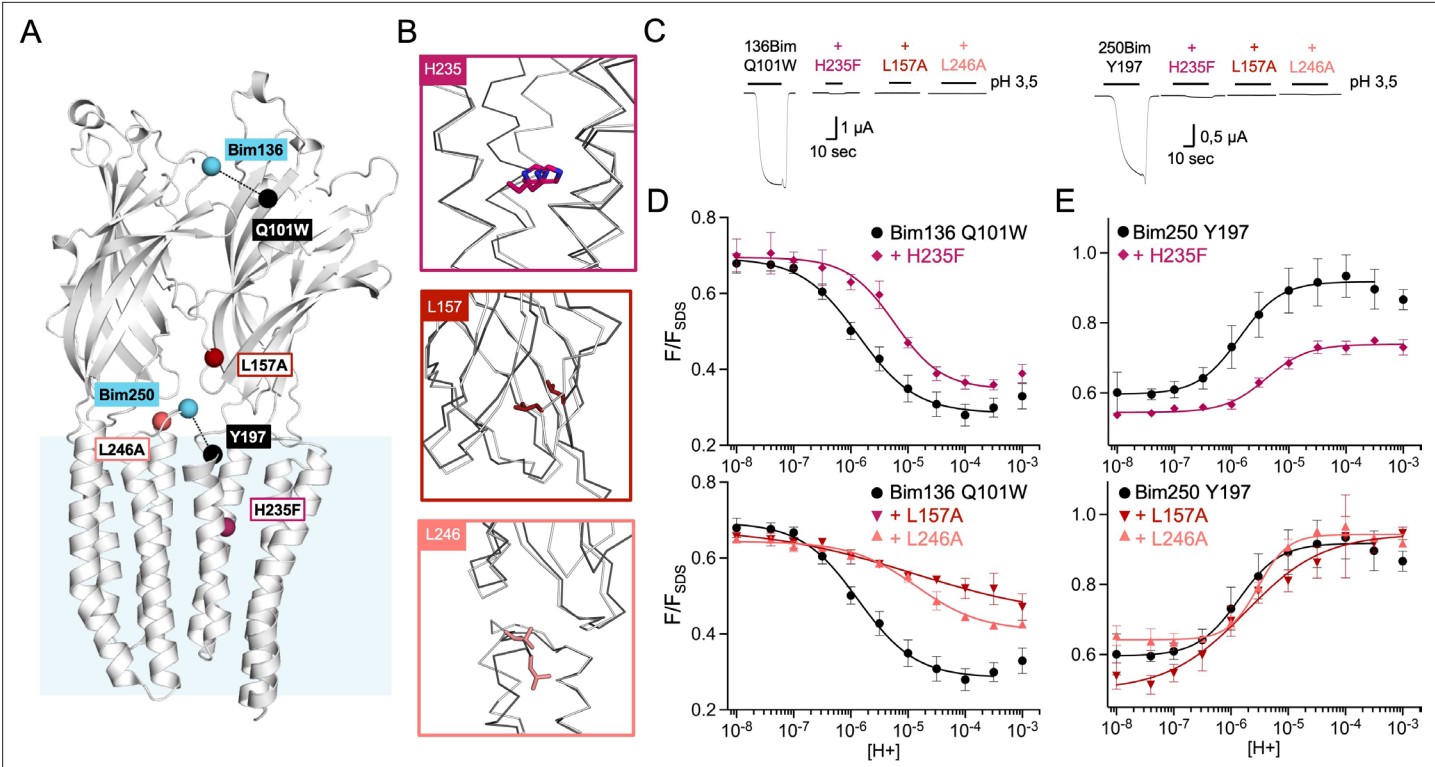

**Figure 4.** Non-functional mutants differentially alter ECD and TMD motions. (**A**) Structure of two monomers of GLIC pH 4 showing the position of the fluorescence sensors (Bim136-Q101W and Bim250-Y197) and three mutations causing a total loss of function. (**B**) Zooms on important re-organizations of the mutated residues between structures at pH 4 (4HFI-gray) and pH 7 (4NPQ-black). (**C**) Electrophysiological recordings in oocytes of the three mutants labeled with bimane showing no current in comparison with GLIC presenting sensor mutations only. pH-dependent curves in fluorescence for the three mutants with the sensor Bim136-Q101W (**D**) and Bim250-Y197 (**E**). ECD, extracellular domain; TMD, transmembrane domain.

The online version of this article includes the following figure supplement(s) for figure 4:

**Figure supplement 1.** Immunofluorescence microscopy data showing GLIC expression at the oocytes surface.

that of Bim136-Q101W-H235Q by more than half a unit (*Figure 5D*). Interestingly, a similar pattern is seen on sensor Bim135-W72 (*Figure 1—figure supplement 1*). Our data thus shows that propofol does act on the global allosteric transitions by displacing the equilibria of both pre-activation and activation. It is noteworthy that propofol is also likely to generate local effects upon binding to modulate the function, which are not investigated here. For instance, its binding into the pore may sterically block ion translocation to produce inhibition (*Fourati et al., 2018*).

## Fit of the data with a three-state MWC model

To characterize the effect of mutations in a more quantitative manner, we fitted the whole data set with a Monod-Wyman-Changeux (MWC) model. Since the fluorescence and electrophysiological pH-dependent curves presented here underlie two major allosteric steps, pre-activation (a fast process causing the changes in fluorescence as previously identified in stopped flow experiments; *Menny et al., 2017*) and activation (a slower process responsible for channel opening), we used a three-state model where the protein is in equilibrium between a resting R state, a pre-active pA state, and an active A state. Changes in fluorescence, although measured here in equilibrium conditions, actually occur with fast kinetics and are likely not related to desensitization. For the measure of activation, we used peak currents recorded in oocytes, assuming that desensitization would be negligible in these conditions. The putative slow-desensitized state of GLIC was thus not included in this model.

In our allosteric model, we first defined a single proton binding site present in five copies with intrinsic affinities for each state named $K_R$, $K_{pA}$, and $K_A$. The equilibria between the states at pH 7 are governed by isomerization constants $L_{pA}=R/pA$ and $L_A=pA/A$ (see Materials and methods for detailed equations). For each fluorescent sensor (Bim136-Q101W and Bim250-Y197), each allosteric state has

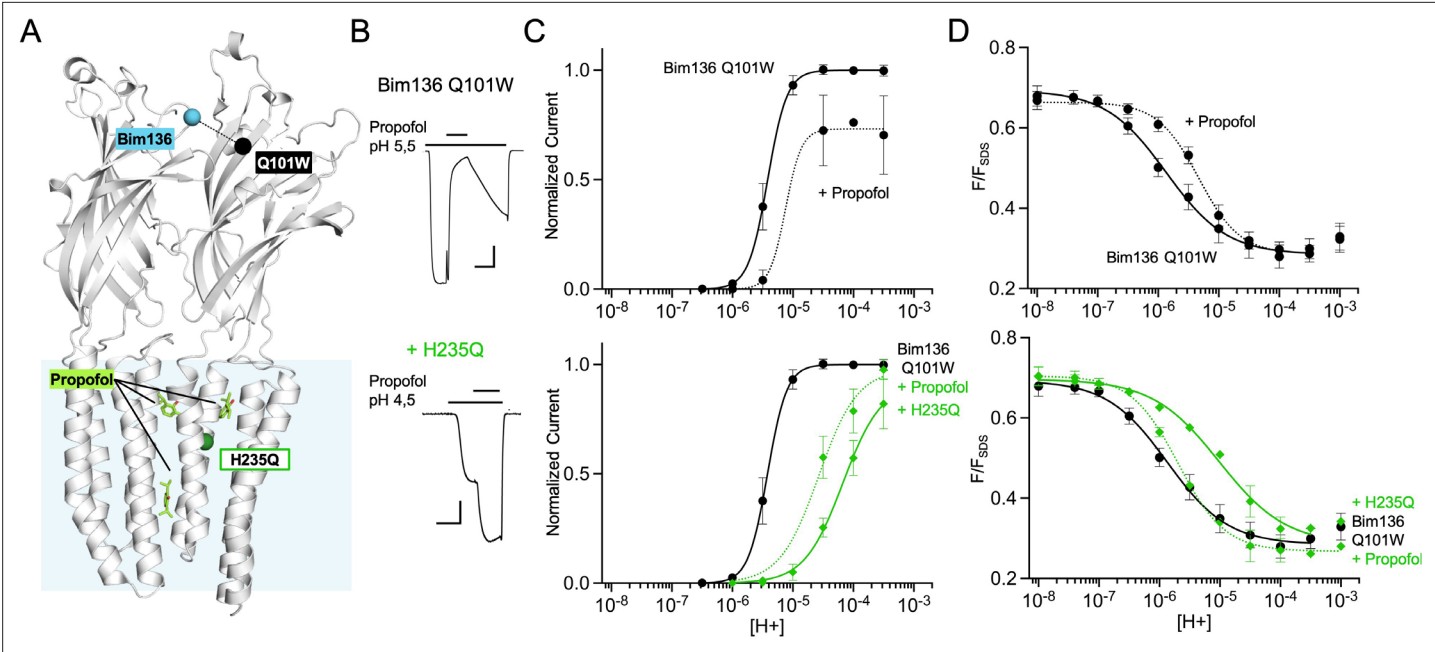

**Figure 5.** Allosteric coupling between the top of the ECD and propofol binding. (**A**) Structure of two monomers of GLIC pH 4 showing positions of the fluorescence sensor Bim136-Q101W at the top of the ECD and three propofol binding sites intra, inter-subunit, and in the pore identified by X-ray crystallography (**Fourati et al., 2018**). (**B**) Example of electrophysiological response to 100 µM propofol during a low pH application (scale bars represent 100 nA and 30 s). (**C**) Electrophysiological pH-dependent curves of Bim136-Q101W with (◆) and without (●) the H235Q mutation showing inhibition and potentiation, respectively. (**D**) Effect of 100 µM propofol on fluorescence quenching without (top panel) and with H235Q mutation (lower panel) for the Bim136-Q101W sensor. ECD, extracellular domain.

a defined fluorescence intensity $F_R$, $F_{pA}$, and $F_A$. As the model involves numerous parameters that cannot be fitted simultaneously given the available data, we adopted a stepwise strategy (summarized in *Figure 6—figure supplement 1*). We formulated several reasonable hypotheses to fit some parameters to the experimental data which were kept fixed while others were constrained to change together:

1. Since the changes in fluorescence occur mainly during pre-activation (*Menny et al., 2017*), we infer that both sensors display identical fluorescence intensity in the pA and A states ($F_{pA}=F_A$).
2. A majority of the allosteric mutants followed with the Bim136-Q101W sensor have almost identical fluorescence values at both ends of the pH curves. This suggests that allosteric transitions at low and high pHs are complete and that at pH 7 a majority of proteins are in the R state ($\overline{R}≈1$, $\overline{R}$ representing the fraction of proteins in the R state) while conversely at pH 4 $\overline{pA} + \overline{A} ≈1$. Consequently, for the Bim136-Q101W sensor, $F_{pH7}=F_R$ and $F_{pH4}=F_{pA}=F_A$.
3. The mutations alter the allosteric isomerization constant between states but not the intrinsic affinities for protons.

## Setting the pre-activation parameters using total loss of function mutants

We started the fitting procedure with the total loss of function mutants, which do not have access to the A state, and simplify the model to one with two states (R and pA). We note that these mutants producing drastic phenotypes, could profoundly alter the protein conformations, possibly including that of the R and pA states and their intrinsic fluorescence. However, the H235F mutant has been shown by X-ray crystallography to adopt a well-folded conformation, captured in the crystal in a 'Locally closed' conformation corresponding to an active-like ECD and resting-like TMD conformation (*Prevost et al., 2012*; *Prevost et al., 2013*) from which we infer that the fluorescence from this mutant reports on WT-like motions.

Fitting of the Bim136-Q101W-H235F curves is constrained by two experimental values (pH50 and $F_{max}$) with three variable parameters, $K_R$, $K_{pA}$, and $L_{pA}$. In consequence, for each value of $L_{pA}$, the two

other parameters are fully constrained by the experimental data. As an illustration, we fixed $L_{pA}$=100. After manual fitting of the curves, we were able to extract $K_R$=3.6×10$^{-6}$ and $K_{pA}$=1.0×10$^{-6}$. We then used these $K_R$ and $K_{pA}$ values to fit the fluorescence quenching curves of Bim250-Y197-H235F, and sensors Bim136-Q101W and Bim250-Y197, only adjusting $L_{pA}$ (*Figure 6—figure supplement 2*). The model thus provides a minimal set of parameters accounting for the pH$_{50}$s and absolute fluorescence changes of these four constructs. It notably suggests that H235F causes a marked stabilization of the R state over the pA state (increase in $L_{pA}$).

## Setting the activation parameters

In a second step, we added the activation state in the MWC model and sought to fit the pH-dependent electrophysiological response curves. Keeping the pre-activation parameters defined above, we found that a three-state model comprising a single proton site could not account for the separation between the fluorescence and electrophysiological curves. With a unique proton site, the model does not allow for more than a fivefold difference, between the curves, when in our experimental data Bim250-Y197, pH$_{50}$s are, respectively, 5.83 and 4.66, more than 1 order of magnitude difference. To fit the activation curves, we thus added a second proton site (named primed, present in five copies), that specifically drives the activation step ($K_{R'}$=$K_{pA'}$, $K_{pA'}$>$K_{A'}$), while the first proton site specifically drives the pre-activation step ($K_R$>$K_{pA}$, $K_{pA}$=$K_A$). This model is reasonable since it is established that several proton sites are contributing to GLIC activation (*Nemecz et al., 2017*). Using this three-state two-site model, we found a set of parameters accounting for the pH-dependent curves of the sensors (*Figure 6A*). With the Bim136-Q101W sensor (*Figure 6C*), variations in the first half of the fluorescence curve result from the apparition of the pA state which is maximally populated (*pA* around 0.5) near the pH$_{50}$ of the fluorescence curve. At lower pHs, the equilibria are further displaced toward the A state, contributing to the decrease in fluorescence in the second half of the curve, and to the parallel apparition of current. With the Bim250-Y197 sensor (*Figure 7*), the mutation introduced causes a destabilization of the A state over the pA state (increase in $L_A$), displacing the pH-dependent activation curve to lower pHs. The pH-dependent fluorescence curve is consequently mainly caused by the apparition of the pA state with a maximal *pA* value reaching more than 0.8.

It should be noted that while the pre-activation parameters are substantially constrained by the experimental data (relying only on the assumption that $L_{pA}$=100 for the H235F mutant), the activation parameters were chosen arbitrarily and other combinations of affinity and isomerization constants could also fit the data. In addition, the data set itself is heterogeneous since fluorescence experiments were performed on purified receptors, while electrophysiology was done on *Xenopus* oocytes. Therefore, the activation parameters used here are only meant to evaluate, as a proxy, the relative effect of each mutation on the activation transition.

## Differential effects of GLIC mutants on pre-activation versus activation

Based on the sensors' parameters, we fitted the various mutants by adjusting the isomerization constants of pre-activation and activation (*Figures 6 and 7*). Overall, reasonable fits can be achieved in most cases. Variations in isomerization constants between mutant and parent sensors were calculated as multiplication factors for pre-activation and activation ($L_{mutant}$/$L_{sensor}$, *Figures 6B and 7A*). In parallel, we performed the whole set of fits with different starting values of the H235F $L_{pA}$ constants (1000 and 100,000), yielding different sets of isomerization constants but in each case similar effects of the mutants (*Figure 6—figure supplement 2—source data 1*, all values presented below were taken from the $L_{pA}$=100 fit unless indicated otherwise). A discrepancy between data points and fits is however consistently observed concerning the apparent cooperativity of most pH-dependent fluorescence curves. The pH-dependent decreases in fluorescence observed experimentally arise over a relatively large range of pHs, while theoretical curves display sharper shapes. It is possible that the pre-activation transition, modeled here by a single allosteric step, might actually involve multiple steps that are not implemented here. Despite this limitation, the model allows us to highlight clear-cut effects.

Experimentally, the most phenotypically striking mutant is Y28F, which produces a 2 orders of magnitude shift of the pH-dependent curves, and a near equalization of fluorescence and electrophysiological pH$_{50}$s. Y28F is readily fitted by the simple assumption that the R-pA equilibrium is strongly displaced toward the R state, that is, the R state is thermodynamically stabilized over both the pA and

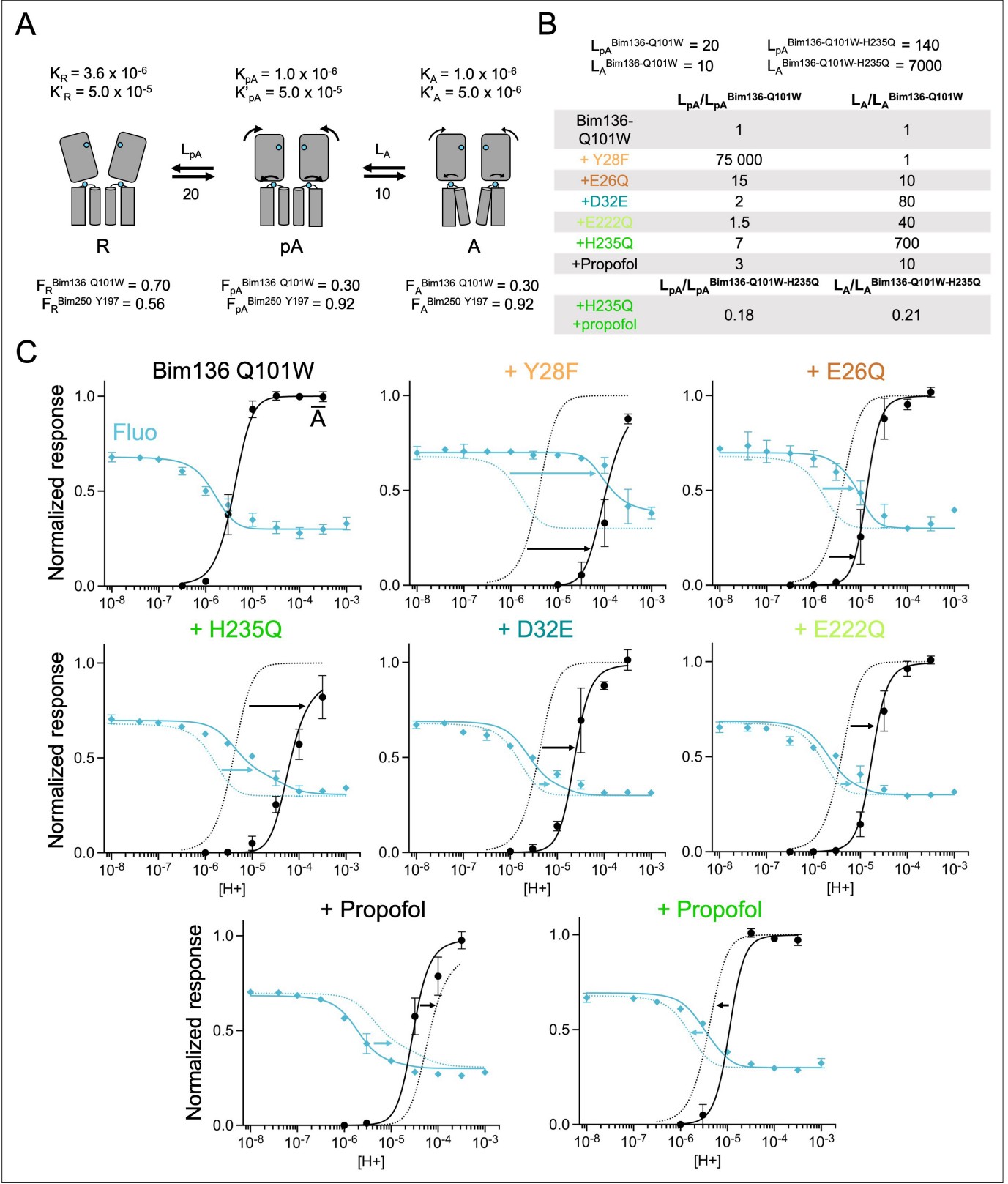

**Figure 6.** The three-state MWC model fits experimental data for Bim136Q101W mutants. (**A**) Scheme showing the three states and parameters of the model. (**B**) Table with multiplication factors of isomerization constants for pre-activation and activation as compared to Bim136-Q101W. For the Bim136-Q101W-H235Q in presence of propofol, the multiplication factors are given in comparison with Bim136-Q101W-H235Q. Isomerization constants of Bim136-Q101W and Bim136-Q101W-H235Q are shown above the table (see full table in *Figure 6—figure supplement 2—source data 1*). (**C**)

*Figure 6 continued on next page*

*Figure 6 continued*

Superposition of experimental data points and theoretical curves. Data points shown as spheres correspond to fluorescence intensities normalized on $F_{SDS}$ (blue, ◆), and to electrophysiological response normalized to the maximal current in (black, ●) except H235Q without propofol for which values were normalized to the values in the presence of propofol. Theoretical curves: the population of A state is shown in black lines and the fluorescence curve (blue line) is calculated from the sum of the three states' fractional populations weighted by their intrinsic fluorescence intensity (see formula in Materials and methods section). For each mutant, the fit from Bim136-Q101W (Bim136-Q101W-H235Q for the last panel) is shown in dotted blue and black lines for a visual comparison and arrows are illustrating the shift in $pH_{50}$. MWC, Monod-Wyman-Changeux.

The online version of this article includes the following source data and figure supplement(s) for figure 6:

**Figure supplement 1.** Workflow for fitting experimental data using the MWC model.

**Figure supplement 2.** A simplified two-state MWC model to fit the total loss of function mutant H235F.

**Figure supplement 2—source data 1.** Summary of isomerization constant set for each mutant.

A states, with no changes in the pA to A equilibrium. In this condition, the fluorescent changes are entirely caused by the apparition of the active state, and the fraction of receptors in the pA state, in these equilibrium conditions, remains below 0.1 % at every pH. This does not mean that this mutant does not populate the pA state during activation, since the pA state may be kinetically favored and actually appear in a transient manner. Mutant E26Q has a milder phenotype, but the fitting also suggests it has a stronger effect on pre-activation than on activation (multiplication factors of isomerization constants of 15 for pre-activation and 10 for activation, *Figure 6B*).

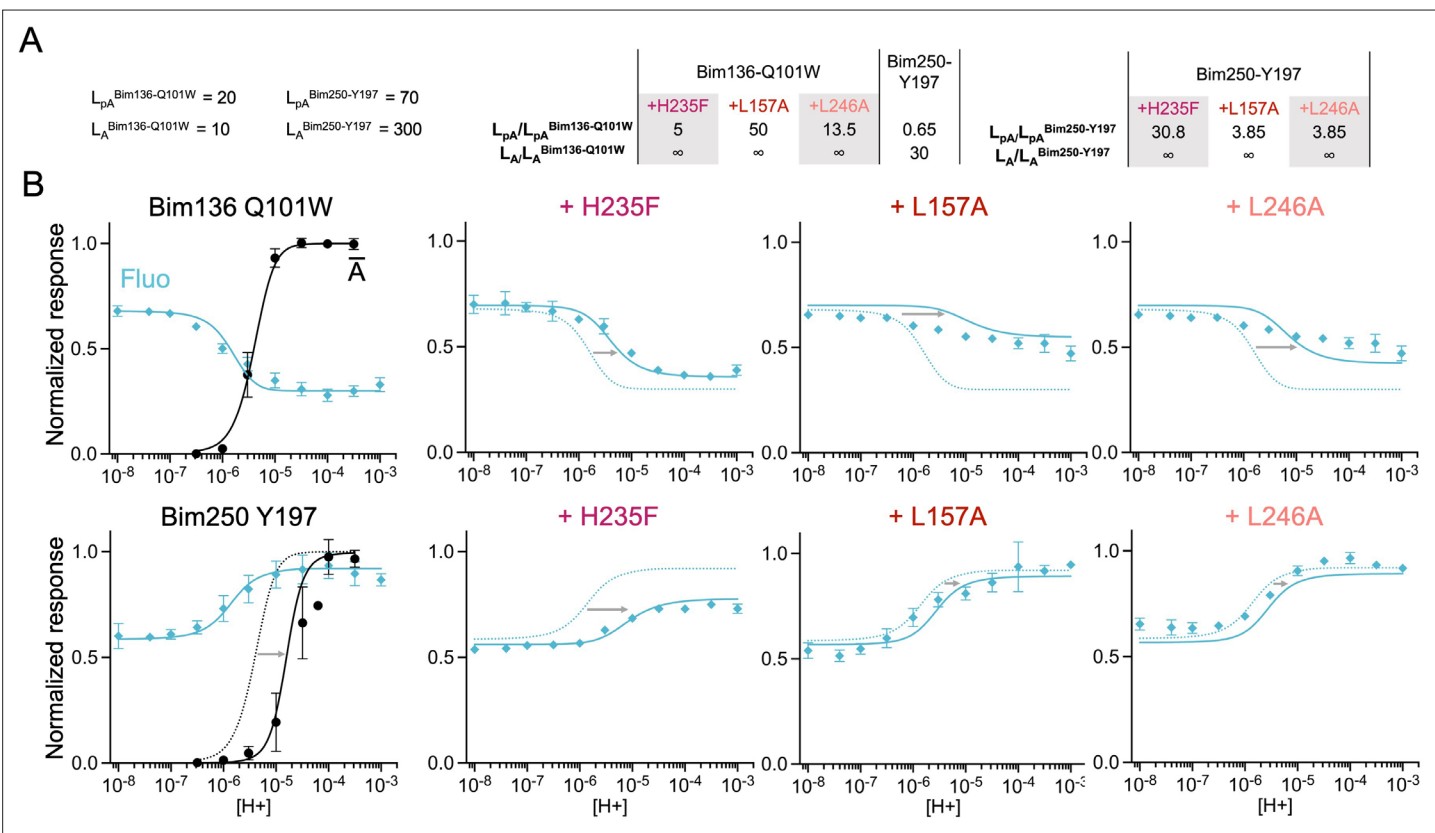

**Figure 7.** The three-state MWC model fits experimental data for total loss of function mutants. Data are presented as in *Figure 6*. (**A**) Multiplication factors of isomerization constants for pre-activation and activation of total loss of function mutants as compared to sensors Bim136-Q101W and Bim250-Y197. (**B**) Superposition of experimental data points and theoretical curves. Data points (in spheres) correspond to fluorescence intensities normalized on $F_{SDS}$ (blue ◆), and to electrophysiological response normalized to the maximal current (black, ●). Theoretical curves: the population of A state is shown in black lines for sensors and the fluorescence curve (blue line) is calculated from the sum of the three states' fractional populations weighted by their intrinsic fluorescence intensity. For each mutant, the fit from the associated sensors (Bim136-Q101W or Bim250 Y197) is shown in dotted lines for a visual comparison and arrows are illustrating the shift in $pH_{50}$. MWC, Monod-Wyman-Changeux.

In contrast, mutants in the lower ECD (D32E), or in the TMD (H235Q and E222Q), are found to preferentially alter the activation transition, destabilizing the A state over the pA state, with small effects on pre-activation (multiplication factor of isomerization constants for pre-activation: 2, 1.5, and 7 and for activation: 80, 40, and 700, respectively). In consequence, they show a large displacement of the activation curve than that of the fluorescence curve when compared to the parent sensor (*Figure 6*). On the Bim136-Q101W sensor and H235Q mutant, propofol acts respectively as a negative and positive allosteric modulator (*Fourati et al., 2018*). For both constructs, propofol is found to have a dual effect, altering principally the activation transition but also the pre-activation transition.

Finally, for total loss of function mutants, while H235F fluorescence quenching could be fitted reasonably well with a two-state R-pA model (*Figure 7*, *Figure 6—figure supplement 2—source data 1*) the best fits of L157A and L246A were of lower quality. In particular, pH-dependent curves of Bim136-Q101W-L157A and Bim136-Q101W-L246A are rather flat, the former being better represented by a straight line. The tentative fits are thus not satisfactory, suggesting that these mutants display complex phenotypes, plausibly driving the conformations into states that are not implemented in our model.

## Investigation of quenching pairs reorganizations using iMODfit and bimane docking

In our previous study, the various quenching pairs were designed on the basis of the comparison of the X-ray structures of GLIC solved at pH 7 and pH 4, selecting pairs of residues that undergo large changes in backbone $C_\alpha$ distances. GLIC-pH 7 is in a non-conductive conformation with a closed hydrophobic gate in the upper part of the pore, consistent with a resting-like state. The GLIC-pH 4 structure shows in contrast an open gate compatible with a conductive conformation (*Cheng and Coalson, 2010*; *Fritsch et al., 2011*; *Sauguet et al., 2013*; *Gonzalez-Gutierrez et al., 2017*) consistent with an active-like structure.

However, the orientation of bimane fluorophore and the surrounding residues including the main quencher is not known for Bim136-Q101W and B250-Y197. Their distances should be taken into consideration to propose a more faithful picture of the underlying molecular reorganizations. In addition, the comparison of crystallographic structures alone does not inform us on the time course of the quenching process during the movement. For instance, at Bim136-Q101W and Bim250-Y197, relatively large changes in $C_\alpha$ distance (2–5 Å) are observed between GLIC-pH 7 and GLIC-pH 4, but one can ask how the distances (and quenching) evolve during these movements.

To investigate these issues, we computed approximate trajectories between the two states using iMODfit, and then modeled on them the bimane/quencher pair using a simple docking approach. iMODfit has been originally designed to fit structures inside electron-microscopy envelopes, notably from a very different starting conformation (*Lopéz-Blanco and Chacón, 2013*). This flexible fitting

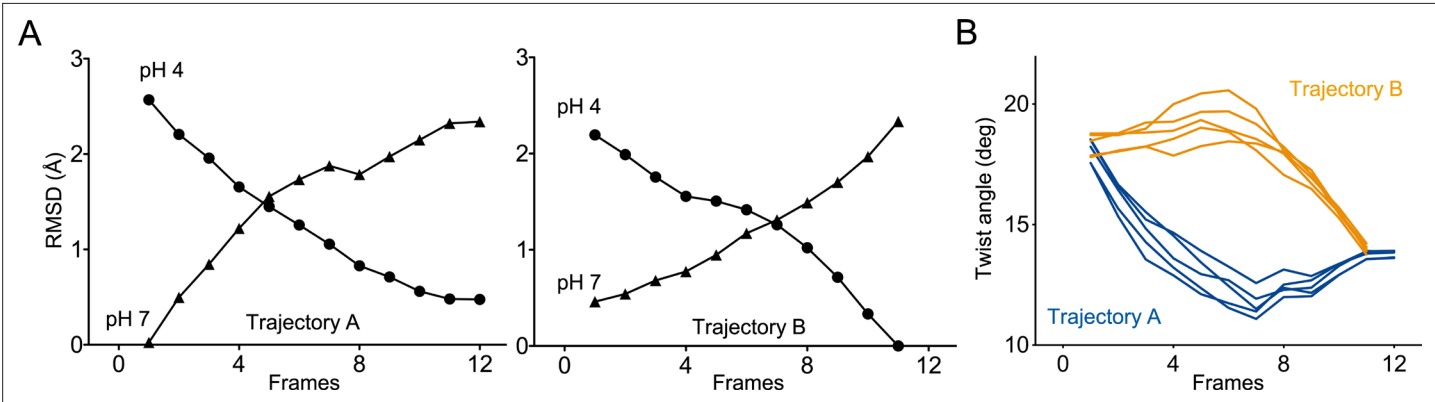

**Figure 8.** Two distinct trajectories for GLIC activation computed using iMODfit. (**A**) RMSD evolution throughout the frames of trajectories A and B against GLIC structures at pH 4 (●); pH 7 (▲), pdb codes are 4HFI and 4NPQ, respectively. Both trajectories are shown with frame one being the closest to GLIC-pH 7. (**B**) Twist angle measured throughout the frames on both trajectories. The twist angle is measured by the angle formed between vectors from the centers of mass of the ECD and TMD as defined in *Calimet et al., 2013*. Each trace corresponds to the trajectory of a single subunit within the pentamer.

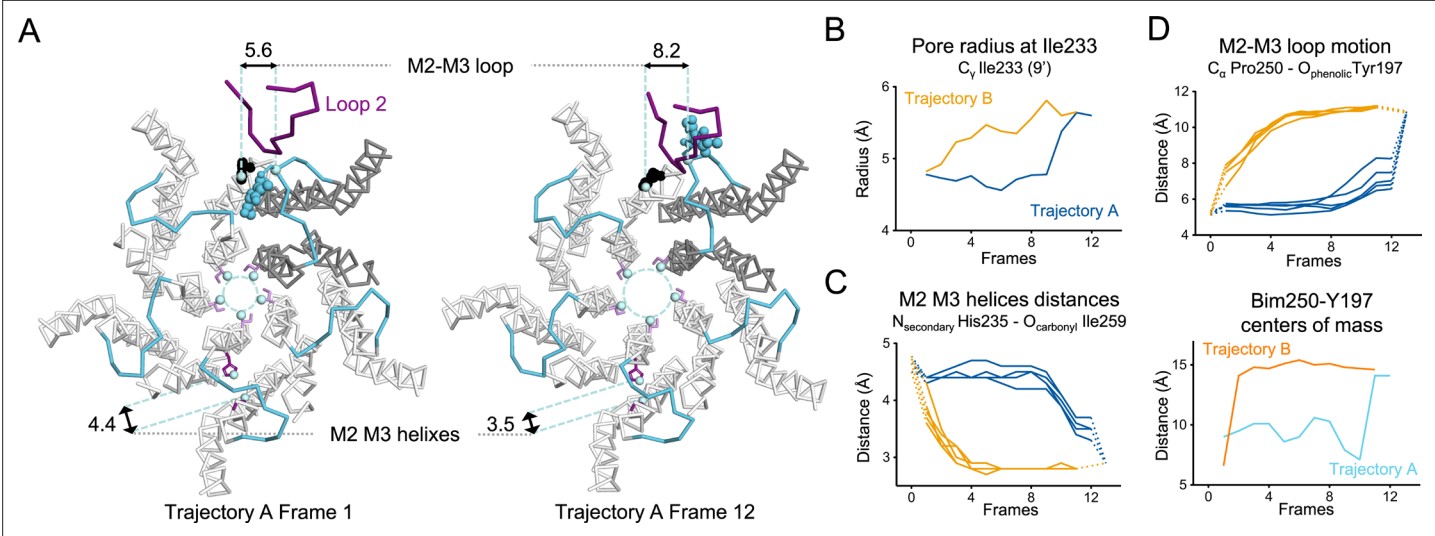

**Figure 9.** Key TMD motions in trajectories A and B. (**A**) Snapshots of GLIC TMD top view in the first and last frame of the trajectory A with a Bim250-Y197 quenching pair modeled at one interface. Bimane is shown in blue and Y197 in black spheres. One subunit is shown in gray, the others are in white, the M2-M3 loop is shown in blue, and the loop 2 from ECD is shown in purple for one subunit. Atoms used for measurements are shown in pale blue spheres and distances are indicated in angstroms. (**B**) Pore radius measured at the Ile233 level. (**C**) Intra-subunit separation of M2 and M3 helices measured between atoms indicated. Points at positions 0 and 13 are the distances measured in pH 4 and pH 7 X-ray structures. (**D**) Inter-subunit distances showing M2-M3 loop outward motion at the Pro250-Tyr197 level (top panel) and between bimane and Tyr197 centroids (bottom panel) in both trajectories A and B. ECD, extracellular domain; TMD, transmembrane domain.

is made via the deformation of the structure using normal mode analysis (NMA) (see Materials and methods section). NMA approximates the surface of the conformational landscape and decomposes the movements into discrete modes. It takes advantage of a simplified but physically meaningful representation of the interaction between the atoms, based on simple springs connecting close pairs of atoms in the native structure. This method provides a time-independent equation and allows the study of slow (biologically relevant) and collective conformational transitions. NMA has been shown previously to allow the study of pLGIC gating mechanisms (*Taly et al., 2005*; *Bahar et al., 2010*). In addition, we have shown on NMDA receptors that iMODfit's NMA-based fitting process can actually visit biologically relevant intermediate structures (*Esmenjaud et al., 2019*). The aim of this study is therefore not to capture the fine details of the transition pathway, but to generate plausible trajectories capturing the main features of the conformational reorganization.

## Generation of two distinct conformational pathways using iMODfit

Two independent trajectories were computed. Trajectory A (12 frames) starts from the closed GLIC-pH 7 structure to reach the open GLIC-pH 4 structure, and trajectory B (11 frames) starts from the GLIC-pH 4 structure to reach the GLIC-pH 7 structure. Both trajectories are fully reversible and are equally relevant to describe either activation or deactivation, since normal modes deformation can be applied in the two directions. RMSD analysis between each frame and the reference structure indicates gradual reorganization of GLIC across the length of both simulations (*Figure 8A*). Both trajectories, when visualized from the resting to active state, show three major reorganizations components: a quaternary twist of the pentamer, a 'central gating reorganization' comprising opening/closure of the pore, and a quaternary compaction of the ECD.

In trajectory A, the twist motion occurs in the first half of the trajectory (*Figure 8B*). This motion describes opposite rotations between ECD and TMD domains, as measured by the twist angle defined by center of mass vectors of the ECD and TMD (*Taly et al., 2005*; *Calimet et al., 2013*). The pore reorganization happens in the second half of the trajectory and leads to the opening of its upper part which contains the activation gate, as measured at Ile233 C$_\alpha$ (also named I9'; *Figure 9A and B*). This motion is associated with a central gating reorganization of the GLIC structure involving: 1/ a tilt of M2 toward M3 (as measured by a decrease in distance between His235 nitrogen and the carbonyl backbone of Ile259; *Figure 9A and C*), 2/ an outward motion of the M2-M3 loop (measured as an

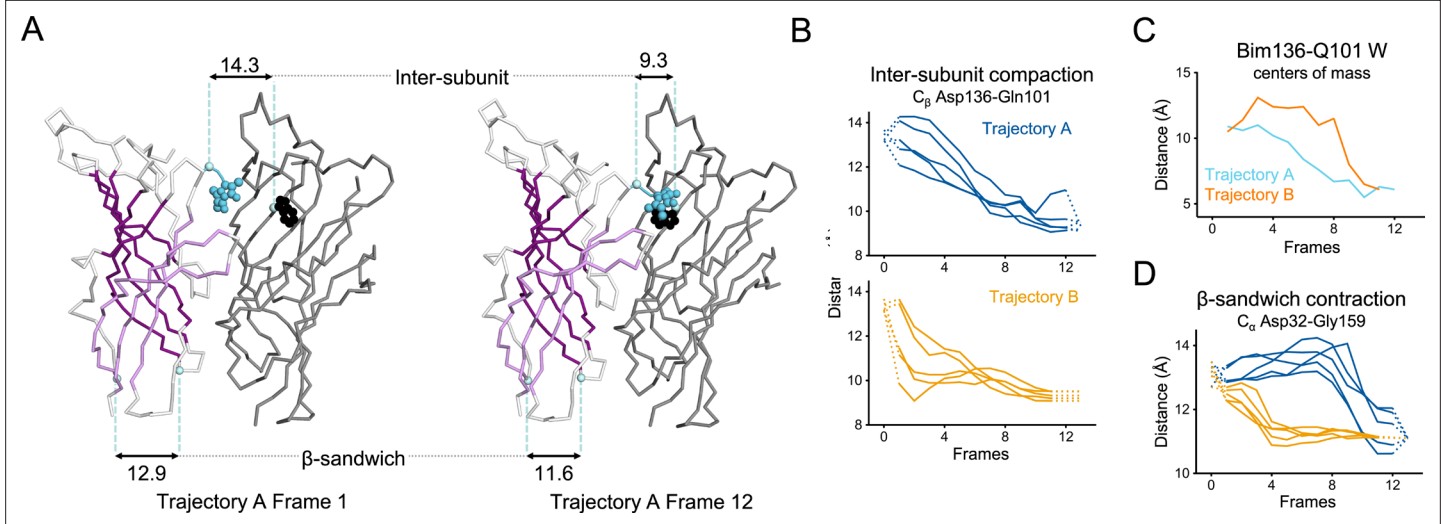

**Figure 10.** Key ECD motions in trajectories A and B. (**A**) Snapshots of two subunits of GLIC ECD in the first and last frame of the trajectory A with a Bim136-Q101W quenching pair modeled at the interface. One subunit is shown in gray, the other in white with sheets of the β-sandwich shown in dark and light purple; bimane is shown in blue and Trp101 in black spheres; $C_\alpha$ and $C_\beta$ atoms used for measurements are shown in pale blue spheres and distances are indicated in angstroms. Inter-subunit distances showing ECD compaction measured at the Asp136-Gln101 level (**B**) and between bimane and Q101W centroids (**C**) in both trajectories A and B. Points at frames 0 and 13 are the distances in pH 4 and pH 7 X-ray structures. (**D**) Intra-subunit distance showing contraction at the bottom of the β-sandwich measured by $C_\alpha$ distances between Asp32 and Gly159. ECD, extracellular domain.

The online version of this article includes the following figure supplement(s) for figure 10:

**Figure supplement 1.** Evolution of ECD inter-subunits distance Arg133-Leu103 at the top of the ECD in iMODfit trajectories.

**Figure supplement 2.** Evolution of ECD inter-subunits distance Lys33-Trp160 at the bottom of the ECD in iMODfit trajectories.

**Figure supplement 3.** Evolution of Bim135-W72 orientation at the ECD intra-subunits in iMODfit trajectories.

increase in distance between Pro250 Cα and the phenolic oxygen of Tyr197; *Figure 9A and D*), and 3/ a contraction of the β-sandwich at the bottom of the ECD (measured as a decrease in distance between $C_\alpha$ of residues Asp32 and Gly159; *Figure 10*). In addition to these two consecutive global motions, the progressive quaternary compaction of the ECD, another crucial landmark of GLIC reorganization, occurs throughout the trajectory. This compaction is quantified through measurement of inter-subunit distances at the top ECD (between $C_\beta$ Asp136/Gln101 and Arg133/Leu103; *Figure 10A, B and C* and *Figure 10—figure supplement 1*), and the bottom ECD (measured by a decrease in the inter-subunit distance of $C_\beta$ Lys33/Trp160, *Figure 10—figure supplement 2*), indicating a progressive decrease in the distance throughout the frames. It is noteworthy that these inter-subunit distances are highly variable, due to the asymmetric nature of the ECDs of the GLIC-pH 7 structure, where each subunit β-sandwich presents a unique orientation as well as relatively high B-factors (*Sauguet et al., 2014*). This variability decreases over the frames to reach the structure of GLIC-pH 4 which is compact and essentially symmetric.

Trajectory B shows substantially the same components but with an inverted sequence of events. The central gating reorganization starts first and is associated with an increase in pore radius at Ile233, followed by the twist motion in the second half of the trajectory, the latter being associated with further fluctuations of the pore radius. The ECD compaction is also spread over the whole trajectory. In conclusion, using iMODfit we could generate two distinct trajectories that are in principle equally plausible to describe a gating transition of GLIC activation or deactivation.

## Visualization of quenching pairs on iMODfit trajectories

To relate the conformational reorganizations of GLIC to our fluorescence quenching data, we modeled the fluorophore/quenching pairs in both trajectories. To this aim, the cysteine and quencher mutations were modeled and the bimane moiety was docked into each frame while keeping it at a covalent-bond compatible distance to the sulfur atom of the cysteine. The distance between bimane centers

of mass and their quenching indole/phenol moieties was then measured in each frame to follow its evolution throughout the trajectories.

For Bim136-Q101W, the procedure shows that Bim136 and the Trp101 indole ring are separated in the resting-like state (first frame of both trajectories), and are in close contact in the active-like state (last frames of both trajectories) (*Figure 10A*). These observations are in good agreement with fluorescence data that show a decrease in fluorescence intensity upon pH drop reporting a decreased distance within the pair. The trajectory A shows a progressive decrease in distance that parallels the ECD quaternary compaction movement ($C_\beta$ Asp136/Gln101). The trajectory B shows a different pattern, characterized by important fluctuations followed by a sharper distance decrease only in the last frames (*Figure 10C*).

For the ECD-TMD interface quenching pair Bim250-Y197, the procedure shows that Bim250 is in close contact with the Tyr197 phenol ring in the resting-like state, while both moieties are separated in the active-like state, the bimane moiety moving on the other side of loop 2 (*Figure 9A*). This is also in agreement with the fluorescence data that showed an increase in fluorescence upon pH-drop indicating that the Bim250 is moving away from its quencher Tyr197. Interestingly, both trajectories A and B show an abrupt change in Bim250-Y197 distances, corresponding respectively to a late versus early separation, and these changes occur during the outward motion of the M2-M3 loop ($P250_{C\alpha}$/ $Y197_O$ distance; *Figure 9D*).

In conclusion, visualizing the quenching pairs using a simple docking procedure shows good agreement with fluorescence. At position P250, data also show a clear switch of bimane from one side of loop 2 to the other during the quenching/dequenching process.

## Discussion

### Long-range allosteric coupling associated with pre-activation and pore-opening processes

In this study, we revisited several fluorescent sensors by performing detailed pH-dependent quenching curves and parallel iMODfit/docking calculations. Our data clearly support that Bim136-Q101W and Bim250-Y197 sensors are *bona fide* reporters of the ECD compaction and the outward M2-M3 motion, respectively. In contrast, data related to the Bim135-W72 sensor (presented and discussed in *Figure 1—figure supplement 1* and *Figure 10—figure supplement 3*) show complex patterns of quenching in both in silico and fluorescence experiments. We infer that, because of the buried location of Bim135 within the protein, it is sensitive to subtle structural reorganizations, the complexity of which precludes clear conclusions. This emphasizes that the fluorescence quenching approach requires screening of multiple positions to select the ones reporting on well-defined local motions.

Using these appropriate sensors, we found that a series of five loss-of-function mutations, which shift the pH-dependent electrophysiological curves to higher concentrations, also shift the pH-dependent fluorescence quenching curve of ECD-compaction at the extracellular top of the protein. The ECD-compaction is thus sensitive to mutations scattered along the protein structure down to the opposite cytoplasmic end, indicating substantial allosteric coupling. Since the conformational motions followed by fluorescence occur early in the pathway of activation, it is expected that a shift in the fluorescence curve will be reflected by a parallel shift in the electrophysiological curve. Mutations in the ECD E26Q and Y28F/C27S both present such a phenotype with similar $\Delta pH_{50}$ in electrophysiology and fluorescence, suggesting that those mutations would mainly impact the pre-activation transition. In contrast, D32E, E222Q, and H235Q lead to a stronger $pH_{50}$ shift in electrophysiology than in fluorescence suggesting that these mutations would alter not only the pre-activation, but also the downstream pore-opening transitions leading to an additive effect on the $pH_{50}$.

### Discriminating pre-activation versus activation phenotypes through allosteric modeling

To interpret the mutant phenotypes in a more quantitative manner, we fitted the whole series of data using a three-state two-site model. We had to implement two proton binding sites to account for the separation of the fluorescence and electrophysiological curves of most constructs. This idea is supported by a mutational analysis that showed that several proton activation sites, located at multiple loci, contribute to activation (*Nemecz et al., 2017*). In addition, chimeric receptors made up

of the GLIC$_{ECD}$ fused to the TMDs of various pLGICs (*Duret et al., 2011*; *Ghosh et al., 2017*; *Laverty et al., 2017*) or of the ELIC$_{ECD}$ fused to the GLIC$_{TMD}$ (*Schmandt et al., 2015*) all preserve a proton-gated ion channel function, with the GLIC$_{ECD}$-GABA $\rho$ $_{TMD}$ chimera showing a markedly biphasic pH-dependent activation curve (*Ghosh et al., 2017*). This suggests that the proton activation sites, whose loci are not known, are scattered throughout the GLIC structure, in both the ECD and the TMD. In our model, we arbitrarily tuned the affinity constants of site 1 to drive the pre-activation transition, and of site 2 to drive the activation transition, to minimize the number of parameters involved.

We also postulated that the various mutants only alter the isomerization constants between states. However, the data set does not allow for the discrimination between effect on binding affinity versus isomerization constants. The effects of mutations on the isomerization constants are thus used here to evaluate the global effect of the mutations on pre-activation versus activation, but it is possible that they actually report on alteration of isomerization constants, affinity constants, or both. Among the various mutations investigated here, E26Q, E222Q, and H235F/Q neutralize the charge of titratable amino acids. It is thus possible that in these cases the mutation eliminates a proton binding site. However, a local impact of a mutation on a proton binding site, or on a set of inter-residues interactions altering the allosteric equilibria, will be equally valid in assigning local structural alterations to pre-active/active phenotypes.

The pattern of effect on L$_{pA}$ versus L$_A$ among the various mutants allows us to dissect their allosteric impact. As anticipated from measured $\Delta$pH$_{50}$, the fits illustrate that Y28F and E26Q principally alter the pre-activation transition and that Bim250, D32E, H235Q, and E222Q principally alter the activation transition, while propofol alters similarly both processes (*Figure 11*). Concerning the total loss of function mutants, we found that they do preserve pre-activation-like allosteric motions, although with an impaired sensitivity and amplitude of the fluorescence curves. H235F is acceptably fitted according to an R-pA model, suggesting that this mutant isomerizes to a pre-active-like state but cannot isomerize further to the active state. L257A and L246A show a more complex phenotype, but fluorescence data show at least partial pre-active-like motions.

## Structural reorganizations associated with pre-activation versus activation

Comparison of the GLIC-pH 7 and GLIC-pH 4 X-ray structure highlighted key reorganizations involved in gating (*Sauguet et al., 2014*), notably a quaternary compaction of the ECD, a tertiary compaction of the β-sandwich in the lower part of the ECD, an outward motion of the M2-M3 loop, and a tilt of the M2 helix toward the M3 helix. Our combined electrophysiological and fluorescence study untangles evaluating the contribution of these specific motions to the pre-activation versus activation transitions.

The ECD quenching pairs at Bim136, Bim133, and Bim33 already showed that pre-activation involves a major quaternary compaction of the whole ECD. We strengthen this idea further by showing that E26Q and Y28F/C27S, that are also located at the subunit interface in the lower part of the ECD, strongly impair the pre-activation process with weaker effects on activation. In addition, the quenching pair at Bim250 showed that the pre-activation involved a key outward movement of the M2-M3 loop. Our data indicate that pre-activation also includes motions of the TMD, since mutation H235Q, as well as propofol binding, are shown to significantly alter pre-activation.

For the activation, our mutational analysis points to a key role of the lower inner part of the ECD β-sandwich (D32E), the M2-M3 loop (Bim250), and the TMD (H235F/Q and E222Q). Interestingly, D32E is involved in strong interactions between sheets in the lower part of the β-sandwich through a salt bridge with R192 (*Figure 3A*). Mutation, D32E, elongating the side chain by one carbon atom is thus predicted to disfavor the β-sandwich compaction. In addition, at the middle of the TMD, H235 from M2 interacts with the main-chain carbonyl of I259 from M3 through an H-bond favoring the interaction between both helices (*Prevost et al., 2012*; *Rienzo et al., 2014*; *Figure 3A*). Its mutation into Q and F is predicted to weaken or abolish this interaction and disfavor the tilt of M2 toward M3. This assumption is consistent with the X-ray structure of the H235F and H235Q mutants which shows a 'locally closed conformation' where M2 and M3 are separated (*Prevost et al., 2012*; *Fourati et al., 2018*). Our data thus provide evidence that the compaction of the β-sandwich and the tilt of M2 are principally involved in the activation process.

The mutational analysis also shows for most mutations mixed effects on the isomerization constants of activation and pre-activation, suggesting that both processes involve overlapping regions. The

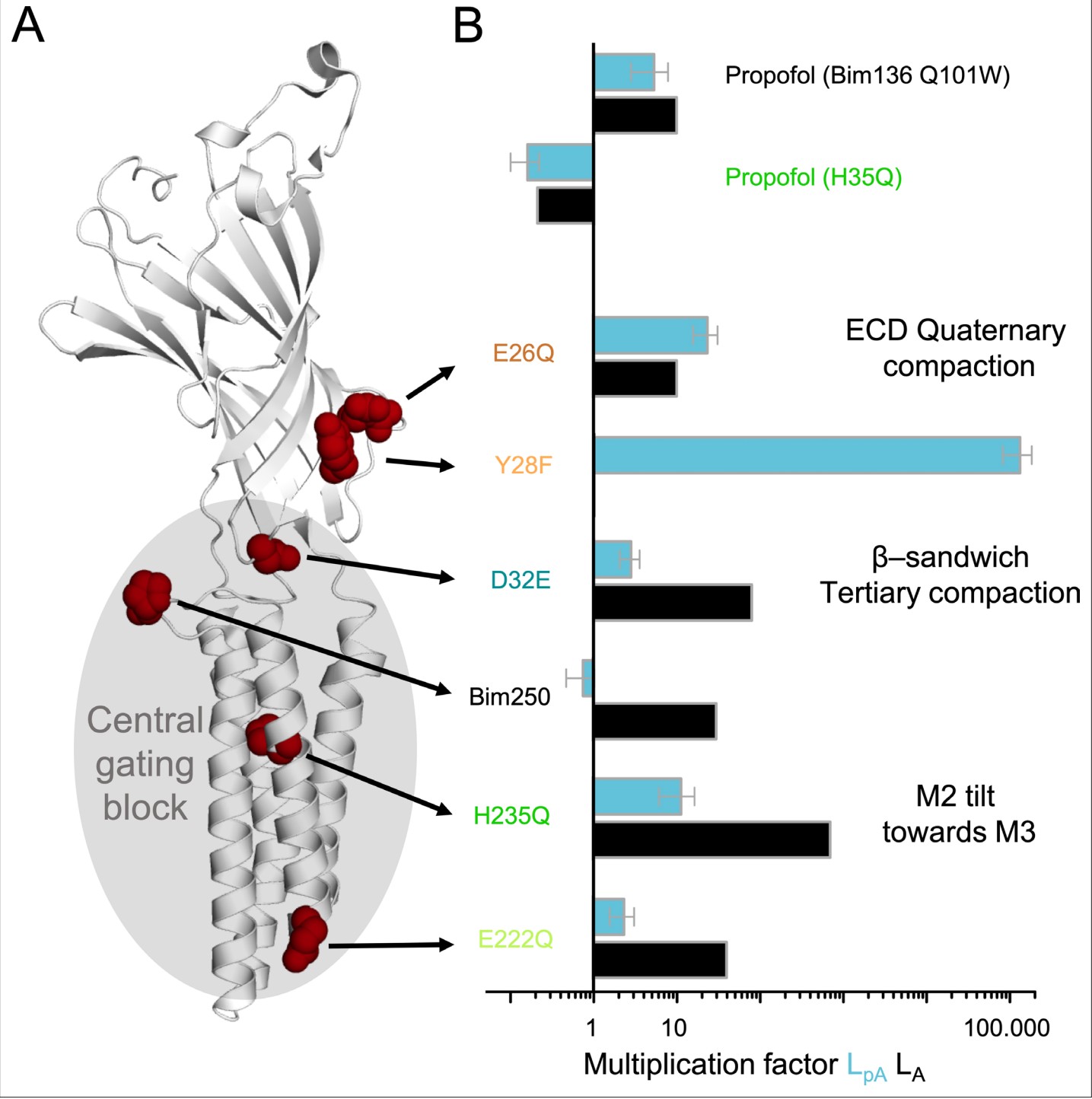

**Figure 11.** Effect of mutations on pre-activation and activation. (**A**) One subunit of GLIC showing the positions of tested mutations in red spheres, with in gray the region involved in the central gating pathway identified by iModFit. (**B**) Multiplication factor shown on a log scale for each mutant to visualize how isomerization constants $L_{pA}$ and $L_A$ were modified in comparison with Bim136-Q101W, or for propofol (H235Q) in comparison with the same mutant without propofol.

Bim250 position is noteworthy in this respect, since the bimane, reporting an outward motion of the M2-M3 loop, monitors pre-activation, while the modification itself (P250C mutation plus reaction with bimane) principally alters the activation process. It is thus plausible that the M2-M3 loop could move in two successive steps, a first one during pre-activation conditioning dequenching and a second one

during activation. In either case, our data further highlight a central role for this loop in ECD-TMD coupling.

## Speculative interpretation of the mutant phenotypes in the context of computational trajectories

The transition pathway of GLIC has been previously studied by atomic-level molecular dynamics simulations in an explicit membrane environment. While the timescale of the transition greatly exceeds that of even the longest possible simulations, two studies addressed this issue. The first one started from the GLIC-pH 4 structure and instantly set it to neutral pH, followed by a 1 μs simulation (*Nury et al., 2010*), yielding concomitant closing of the pore and twist of the whole structure. The second one is based on the string method, using the 'swarms of trajectories' approach, computing a trajectory between GLIC-pH 7 and GLIC-pH 4 (*Lev et al., 2017*). The trajectory shows a sequence of events starting from the closed to the open conformation. A first major reorganization involves the opening of the pore, its hydration, and the compaction of the lower part of the ECD β-sandwich. This is followed by a major reorganization of the ECD, notably its twist and its quaternary compaction. This sequence of events appears hardly compatible with our quenching data, although a comprehensive integration of both sets of data would require extensive in silico investigations of the bimane-labeled mutants to analyze the reorganizations of the quenching pairs. Of note, an important limitation of the method is that it implicitly postulates the occurrence of a single trajectory. However, a coarse-grained simulation (hybrid elastic-network Brownian dynamics) predicted two possible pathways for GLIC gating, that are characterized by different compactions of the ECD (*Orellana et al., 2016*).

In this study, we performed iMODfit/bimane docking calculations and generated two distinct trajectories with an inverted sequence of events. While these trajectories are coarse and do not implement fine atomistic interactions, they allow the visualization of plausible collective motions in relation to the reorganization of the quenching pairs.

Remarkably, both trajectories show complex quaternary asymmetric reorganizations of ECD compaction. As stated above, ECD compaction is critically involved in pre-activation, a feature consistent with recent work by electron paramagnetic resonance (EPR) spectroscopy (*Tiwari et al., 2020*) showing a proton-induced inward tilting motion of the ECDs, and recent cryoEM work showing marked structural flexibility of the ECD in the closed-channel state at pH 7 (*Rovsnik et al., 2021*). Interestingly, fluorescence curves of pre-activation, especially the one of Bim136-Q101W, are endowed with markedly low cooperativities. We thus speculate that this low cooperativity might arise from the contribution of multiple asymmetric intermediate states to the transition, in a manner reminiscent of the asymmetric motions recently described for the desensitization of the GABA$_A$ receptor (*Gielen et al., 2020*).

In addition, both trajectories show a 'central gating motion' involving several key concerted reorganizations: a compaction of the lower part of the β-sandwich, an outward movement of the M2-M3 loop, as well as a tilt of the M2 helix toward M3, that involves a marked increase in the opening of the pore. Similar structural couplings are also observed in string simulations (*Lev et al., 2017*). This observation nicely parallels our finding that these motions are principally involved in activation. We can thus speculate that the central gating motion constitutes the heart of the activation transition.

Concerning the order in which reorganizations are observed, trajectory A is a better fit to the fluorescence data. It suggests a scenario involving, during pre-activation, progressive ECD compaction and beginning of the M2-M3 loop motion, generating the fluorescence variations. Then the M2-M3 loop completes its movement in concert with β-sandwich compaction and pore opening. Future computational studies are needed to explore this possibility.

## Consequences on the gating mechanism within the pLGIC family

The conservation of the general gating mechanism between bacterial and eukaryotic pLGICs is well documented by the available structures with the common allosteric regulatory sites for ligands and mutations (*Sauguet et al., 2015*; *Bertozzi et al., 2016*; *Rienzo et al., 2016*), together with the allosteric compatibility between eukaryotic and prokaryotic ECD/TMD domains to form functional chimeras (*Duret et al., 2011*; *Moraga-Cid et al., 2015*; *Laverty et al., 2017*). It is therefore tempting to speculate that the pre-activation transition of GLIC that we characterize here might have counterparts in human neurotransmitter-gated receptors. In this line, some recent structures of eukaryotic

receptors including the 5-HT$_3$R (*Polovinkin et al., 2018*), the GABA$_A$R (*Masiulis et al., 2019*) and the GlyR (*Yu et al., 2021*) show pre-active-like conformations characterized by marked agonist-elicited reorganization of the ECD but a closed channel at the TMD. Additionally, the flipped or primed states, where the conformational change of the orthosteric site is predicted to be complete, but where the channel is closed, would fit the functional requirement of a pre-active state (*Lape et al., 2008*; *Plested, 2014*).

Our work also investigates the mechanism of action of allosteric mutations by measuring their effects at different levels of the protein, dissecting their phenotype along the gating pathway (*Galzi et al., 1996*). Allosteric mutations of neurotransmitter-gated receptors, causing congenital pathologies including myasthenia and hyperekplexia have been extensively studied (*Taly and Changeux, 2008*; *Bode and Lynch, 2014*; *Hernandez and Macdonald, 2019*). Most of the hot spots mutated here on GLIC were found associated with pathologies on human receptors. In particular, the lower part of the ECD-ECD interface is the site of a de novo S76R mutation in GABA$_A$ α1 (homologous to Glu26) causing epilepsy (*Johannesen, 2016*) and the mutation L42P in the nAChR δ (homologous to Cys27) causing myasthenia (*Shen et al., 2008*). This latter mutation (as well as mutation of N41, homologous to E26) decreases activation kinetics and this residue was shown to be energetically coupled to Y127 on the other side of the interface. Interestingly, equivalent residues in GLIC (C27, E26, and Y111) are part of a water network at the bottom of the ECD (*Figure 2A*). Another noteworthy example is the mutation P250T in GlyR α1 that causes hyperekplexia (*Saul et al., 1999*) and which is homologous to E222 in GLIC. Interestingly, on the glycine receptor α1, other mutations have been studied by single-channel recordings and are described to affect principally a flip pre-activation-like step for A52S in the loop 2 at the ECD-ECD interface (*Plested et al., 2007*) or gating for K276E on the M2-M3 loop (*Lape et al., 2012*). These data suggest that mutations produce similar allosteric perturbations on GLIC and GlyR in those regions.

Our work on GLIC provides general mechanisms of how mutations affect pLGICs transitions and further documents conformational changes, beyond information provided by structures. Further work, for example by voltage-clamp fluorometry, would be required to challenge such mechanisms in the context of congenital pathologies on neurotransmitter receptors.

# Materials and methods

**Key resources table**

| Reagent type (species) or resource | Designation | Source or reference | Identifiers | Additional information |
|---|---|---|---|---|
| Gene (*Gloeobacter violaceus*) | *glvl*, GLIC | UniProt | Q7NDN8 | |
| Strain, strain background (*Escherichia coli*) | BL21(DE3) C43 | Sigma-Aldrich | CMC0019 | Chemically competent cells |
| Biological sample (*Xenopus laevis*) | *Xenopus* oocytes | Centre de Ressources Biologiques *Xénopes* (Rennes- France) and Ecocyte Bioscience (Dortmund-Germany) | | |
| Antibody | Anti-HA Tag (rabbit) | Euromedex | HA-1A1-20 µL | (1:200) |
| Antibody | Anti-rabbit – Alexa Fluor 645 (goat) | Molecular probes | A21246 | (1:1000) |
| Recombinant DNA reagent | Pet20b-MBP-GLIC | *Bocquet et al., 2007* | | |
| Recombinant DNA reagent | pMT3-GLIC-HAtag | *Nury et al., 2011* | | |
| Recombinant DNA reagent | Pmt3-GFP | *Nury et al., 2011* | | |
| Chemical compound, drug | Monobromo-Bimane | Thermo Fisher Scientific | M1378 | |
| Chemical compound, drug | Bunte salt Bimane | *Menny et al., 2017* | | |

*Continued on next page*

*Continued*

| Reagent type (species) or resource | Designation | Source or reference | Identifiers | Additional information |
|---|---|---|---|---|
| Chemical compound, drug | Propofol | Sigma-Aldrich | Y0000016 | |
| Software algorithm | iMODfit | *Lopéz-Blanco and Chacón, 2013* | | |
| Software algorithm | MOLEonline | *Pravda et al., 2018* | | |
| Software algorithm | Clampfit | Molecular devices | | |
| Software algorithm | AxoGraph X | | https://axograph.com/ | |

## Mutagenesis

All GLIC mutants were obtained using site-directed mutagenesis on the C27S background of GLIC, except Bim136-Q101W-Y28F (C27) for which the endogenous cysteine was introduced back. Similarly to previous studies (*Sauguet et al., 2014*; *Menny et al., 2017*; *Nemecz et al., 2017*), two different vectors were used: a pet20b vector with GLIC fused to MBP by a linker containing a thrombin cleavage site under a T7 promoter for expression in *Escherichia coli* BL21; a pmt3 vector for expression in oocytes with GLIC containing a Cter HA tag and in Nter the peptide signal from α7-nAChR. Incorporation of the mutations in both vectors was verified by sequencing.

## GLIC mutants production and purification

Protein production of MBP-GLIC and labeling was done as previously described (*Menny et al., 2017*) with a few modifications. In brief, MBP-GLIC was expressed in BL21 *E. coli* cells overnight at 20 °C after induction by 100 μM IPTG. Cells were collected and resuspended in buffer A containing 20 mM Tris; 300 mM NaCl at pH 7.4 and subsequently disrupted by sonication. After membrane separation by ultracentrifugation, membrane proteins were extracted overnight in buffer A supplemented with 2 % DDM. After ultracentrifugation, supernatant was incubated with amylose resin and MBP-GLIC was eluted using buffer A supplemented with DDM 0.02 % and a saturating concentration of maltose. To remove the endogenous maltoporin contaminant, a first size exclusion chromatography was performed on superose 6 10/300 GL in buffer A with 0.02 % DDM. GLIC-MBP concentration was measured and the protein was incubated overnight at 4 °C with thrombin to cleave off MBP and with monobromobimane (mBBr) at a 1:5 (GLIC monomer:fluorophore) ratio, to label the protein. The mBBr dye being solubilized in DMSO, the sample volume was adjusted to remain below 1 % DMSO final concentration. After labeling, a second gel filtration was done to get rid of the MBP and unbound dye molecules. GLIC-Bimane samples were flash-frozen in liquid nitrogen and stored at –80 °C prior to fluorescence measurements.

## Steady-state fluorescence measurements

Fluorescence measurements were done as previously described (*Menny et al., 2017*). Samples were equilibrated to room temperature and diluted with buffer A with 0.02 % DDM to reach a concentration around 40 μg.ml$^{-1}$. Fluorescence recording buffers consisting of 300 mM NaCl, 2.7 mM KCl, 5.3 mM $Na_2HPO_4$, and 1.5 mM $KH_2PO_4$ were prepared beforehand and their pH was adjusted either to 7.4 or to different pH in order to reach the desired pH value (from pH 8 to 3) after mixing equal volumes with buffer A 0.02 % DDM. Measurements were done at 20 °C in 1 ml disposable UV transparent 2.5 ml cuvettes in a Jasco 8200 fluorimeter with 385 nm excitation wavelength and the emission spectra were recorded through 2.5 nm slits from 420 to 530 nm. Parameters were kept constant throughout the study. On the sample at pH 7.4, an addition of SDS to reach 1 % final concentration was done to obtain the $F_{SDS}$ value and a tryptophan emission spectrum was done before and after SDS addition in order to monitor denaturation.

Fitting of fluorescence measurements was done on each fluorescence series (values from 1 pH range) with at least three series per mutant using the following Hill equation:

$$y\left(x\right) = \frac{\Delta F_{max} + x^{n_H}}{x^{n_H} + EC_{50}^{n_H}} + F_0$$

where $\Delta F_{max}$ represents the maximal change in fluorescence amplitude, $F_0$ represents the initial fluorescence at pH 7.8; $n_H$ represents the Hill number, and $EC_{50}$ represents the proton concentration for which half of the maximal fluorescence change is measured. For Bim136-Q101W and Bim250-Y197 and in some other mutants, we excluded from the fit the data point below pH 3.5 that show a small but significant change in fluorescence intensity in the opposite direction to the quenching curves. We did not fit the Bim135-W72 mutant that shows a bell-shaped curve.

## Electrophysiological recordings

Electrophysiological recordings of GLIC were made on *Xenopus* oocytes provided either by the Centre de Ressources Biologiques Xénopes (Rennes-France) or by Ecocyte Bioscience (Dortmund-Germany). Recordings were made as previously described (*Nury et al., 2011*) with oocytes 48–96 hr post nucleus injection with a mix containing 80 ng.µl$^{-1}$ of GLIC cDNA and 25 ng.µl$^{-1}$ of GFP cDNA. Recordings were done in MES buffer containing 100 mM NaCl, 3 mM KCl, 1 mM CaCl2, 1 mM MgCl2, and 10 mM MES with pH adjusted by addition of 2 M HCl. The perfusion chamber contained two compartments and only a portion of the oocyte was perfused with low pH solution. Bunte salt bimane labeling was performed prior to recording by incubation for 1 hr at room temperature with the dye concentrated at 1 mM in MES buffer. To correct data for rundown, a solution with a pH value in the middle of the pH range (usually pH 5) was used as a reference at the beginning and the end of the recording and every 3/4 applications. To limit the effect of propofol that can stay in the membrane in-between applications (*Heusser et al., 2018*), only a limited number of pH solutions were tested per oocyte.

Electrophysiological recordings were analyzed using AxoGraph X and Prism was used to fit individual pH-dependent recording using the Hill equation:

$$y\left(x\right) = \frac{I_{max} + x^{n_H}}{x^{n_H} + EC_{50}^{n_H}}$$

where $I_{max}$ represents the maximal current in percentage of the response from the reference solution. $n_H$ represents the hill number and $EC_{50}$ represents the proton concentration for which half of the maximal electrophysiological response is recorded.

## *Xenopus* oocytes immunolabeling

Mutants generating currents smaller than 500 nA at high proton concentrations (pH 3.5) were categorized as non-functional. For these non-functional mutants, expression tests were performed by immunolabeling of oocytes as previously described (*Prevost et al., 2012*; *Sauguet et al., 2014*). 34 days postinjection, GFP-positive oocytes were fixed overnight in paraformaldehyde (PFA) 4 % at 4 °C. Immunolabeling was performed after 30 min saturation by 10 % horse serum in phosphate-buffered saline. Rabbit anti-HA-tag primary antibody was incubated for 90 min in 2 % horse serum and the secondary antibody anti-rabbit coupled to Alexa Fluor 645 was incubated for 30 min. After a second PFA fixation overnight, oocytes were included in warm 3 % low-melting agarose and 40 µm slices were made using a vibratome on a portion of the oocyte. Several slices per oocyte were mounted on a slide and analyzed in an epifluorescence microscope using constant exposure time between non-functional mutant and functional mutants used as positive controls.

## Molecular modeling

The iMODfit flexible fitting method (*Lopéz-Blanco and Chacón, 2013*) searches the conformational space using the lowest normal modes for the best cross-correlation fit of a starting conformation atomic model into a target conformation density map. Two trajectories were generated here. In trajectory A structure, 4NPQ (GLIC-pH 7) is fitted to the density of 4HFI (GLIC-pH 4), and in trajectory B structure, 4HFI is fitted to the density of 4NPQ.

The detailed procedure is performed as follows, taking as an example trajectory A: 1/ A computed EM density map was generated for the X-ray structure of the target 4NPQ using the pdb2vol tool (called 4NPQ map). The EM density map resolution was set to 5 Å and the grid size to 0.5 Å, that is, the resolution was set at a relatively large value to avoid being locked in local minima during the iMODfit procedure.

2 / 4 HFI was represented with the detailed all heavy-atoms force field (all atoms are considered except hydrogens), called the 4HFI model.

3/ The lowest-frequency NMA-modes of the 4HFI model were computed. For the subsequent steps, the range of modes considered (−n option) was set to 0.5, that is, half of the modes, corresponding to the lower frequency modes, are considered for computing the conformational changes.

4/ During the iMODfit procedure, starting from the 4HFI model, 10 % of the modes are randomly selected and used to generate a very small conformational change. The new conformation is used to compute a simulated density map, and the new conformation is accepted only if the cross-correlation between the targeted 4NPQ and simulated maps improves. This process is repeated iteratively until the conformation deviates by an RMSD of 0.5 Å from the starting/previous model, in which case an intermediate structure is generated and stored. The entire process is then repeated iteratively to generate a series of intermediate states that progressively converge to the targeted structure.

The geometry of the ion channel has been computed with MOLEonline webserver (mole.upol.cz), with the 'pore' mode (*Sehnal et al., 2013*). We used the FreeRadius value computed at the level of the I9' residue.

For the Bimane docking procedure on each intermediate structure, the position of side chains was first optimized with the software Scwrl4 (*Krivov et al., 2009*) while keeping the main chain rigid. This step also allowed the introduction of point mutations. The structure of the protein and bimane was converted to pdbqt files with the software open babel 2.4.1. Covalent docking was then performed with the software smina (*Koes et al., 2013*). The box for docking has been defined around the mutated cysteine residue, with a size of 30 Å in each direction. Covalent docking forced the bimane to be in appropriate distance with the sulfur atom of the introduced cysteine. Only the first pose was kept for further analysis.

## MWC model building

To build a three-state MWC model, the following equations were used to obtain the population of each state resting, pre-active and active:

$$\overline{A} = \frac{(1+\alpha)^5 \times (1+\alpha')^5}{(1+\alpha)^5 \times (1+\alpha')^5 + L_{pA}L_A\left(1+C_{pA}C_A\alpha\right)^5 \times \left(1+C'_{pA}C'_A\alpha'\right)^5 + L_A\left(1+C_A\alpha\right)^5 \times \left(1+C'_A\alpha'\right)^5}$$

$$\overline{pA} = \frac{L_A\left(1+C_A\alpha\right)^5 \times \left(1+C'_A\alpha'\right)^5}{(1+\alpha)^5 \times (1+\alpha')^5 + L_{pA}L_A\left(1+C_{pA}C_A\alpha\right)^5 \times \left(1+C'_{pA}C'_A\alpha'\right)^5 + L_A\left(1+C_A\alpha\right)^5 \times \left(1+C'_A\alpha'\right)^5}$$

$$\overline{R} = \frac{L_{pA}L_A\left(1+C_{pA}C_A\alpha\right)^5 \times \left(1+C'_{pA}C'_A\alpha'\right)^5}{(1+\alpha)^5 \times (1+\alpha')^5 + L_{pA}L_A\left(1+C_{pA}C_A\alpha\right)^5 \times \left(1+C'_{pA}C'_A\alpha'\right)^5 + L_A\left(1+C_A\alpha\right)^5 \times \left(1+C'_A\alpha'\right)^5}$$

With constants defined below:

$$L_{pA} = \frac{\overline{R_{pH8}}}{pA_{pH8}} \quad C_{pA} = \frac{K_{pA}}{K_R} \quad C'_{pA} = \frac{K'_{pA}}{K'_R} \quad \alpha = \frac{[H^+]}{K_A}$$

$$L_A = \frac{\overline{pA_{pH8}}}{A_{pH8}} \quad C_A = \frac{K_A}{K_{pA}} \quad C'_A = \frac{K'_A}{K'_{pA}} \quad \alpha' = \frac{[H^+]}{K'_A}$$

The weighted fluorescence value was calculated as followed:

$$F = \overline{R} \times F_R + \overline{pA} \times F_{pA} + \overline{A} \times F_A$$

With fluorescence values set at:

$$F_R^{Bim136-Q101W} = 0.70 \quad F_R^{Bim250-Y197} = 0.56$$

$$F_{pA}^{Bim136-Q101W} = 0.30 \quad F_{pA}^{Bim250-Y197} = 0.92$$

$$F_A^{Bim136-Q101W} = 0.30 \quad F_A^{Bim250-Y197} = 0.92$$

Isomerization constants were manually adjusted to fit theoretical and experimental fluorescence quenching curves and normalized electrophysiological curves. Of note, the fluorescence variations of E26Q mutant were normalized to that of the Bim136-Q101W, to correct for its effect on the fluorescence at pH 7 which likely reflects an alteration of the structure of the resting state, independently of

the allosteric transitions. Additionally, the $A$ population for the Y28F mutant does not reach 1, so it was normalized in order to compare the values with the normalized experimental data.

## Acknowledgements

The work was supported by the 'Agence Nationale de la Recherche' (Grant ANR-13-BSV8-0020, Pentagate), the doctoral school ED3C and the 'Foundation pour la Recherche Médicale' (PhD funding to SNL), the 'Initiative d'Excellence' (cluster of excellence LABEX Dynamo, ANR-11-LABX-0011 to AT) and the ERC (Grant no. 788974, Dynacotine). The authors would like to thank Stuart Edelstein for helping with MWC equations, Marc Gielen, Akos Nemecz, and Marie Prévost for critical reading of the manuscript.

## Additional information

### Funding

| Funder | Grant reference number | Author |
|---|---|---|
| Agence Nationale de la Recherche | ANR-13-BSV-0020 | Solène N Lefebvre<br>Anaïs Menny<br>Karima Medjebeur<br>Pierre-Jean Corringer |
| Agence Nationale de la Recherche | ANR-11-LABX-0011 | Antoine Taly |
| European Research Council | grant No. 788974 | Pierre-Jean Corringer |
| Sorbonne University - Doctoral school ED3C | PhD fellowship | Solène N Lefebvre |
| Fondation pour la Recherche Médicale | PhD fellowship complement | Solène N Lefebvre |

The funders had no role in study design, data collection and interpretation, or the decision to submit the work for publication.

### Author contributions

Solène N Lefebvre, Conceptualization, Formal analysis, Investigation, Methodology, Validation, Visualization, Writing – original draft, Writing – review and editing; Antoine Taly, Conceptualization, Investigation, Methodology, Resources, Validation, Writing – review and editing; Anaïs Menny, Investigation, Methodology, Validation, Writing – review and editing; Karima Medjebeur, Investigation, Validation; Pierre-Jean Corringer, Conceptualization, Funding acquisition, Project administration, Supervision, Writing – original draft, Writing – review and editing

### Author ORCIDs

Solène N Lefebvre http://orcid.org/0000-0002-1333-2042
Antoine Taly http://orcid.org/0000-0001-5109-0091
Anaïs Menny http://orcid.org/0000-0002-6044-4119
Pierre-Jean Corringer http://orcid.org/0000-0002-4770-430X

### Decision letter and Author response

Decision letter https://doi.org/10.7554/eLife.60682.sa1
Author response https://doi.org/10.7554/eLife.60682.sa2

## Additional files

### Supplementary files
• Transparent reporting form

## Data availability

Table 1 included in the manuscript correspond to a summary table for figures 4 to 8.

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
