## [Decision Letter]

**Acceptance summary:**

How structural motions in pentameric ligand gated ion channels(pLGICs) lead to functional channel gating transitions remains poorly understood. Using the prototypic bacterial proton-gated channel GLIC, fluorescent reporters of protein conformation and kinetic modeling, the authors found a set of mutations that mostly alter pre-gating transitions and others that mainly alter gating (pore opening). Using structural trajectories identified by normal mode analysis to interpret their data in structural terms suggests that pre-activation transitions involve quaternary compaction of the extracellular domain and that activation involves a re-organization of a central gating region. This paper adds new mechanistic information about pLGIC activation.

**Decision letter after peer review:**

Thank you for submitting your article "Mutational analysis to explore long-range allosteric coupling and decoupling in a pentameric channel receptor" for consideration by *eLife*. Your article has been reviewed by 2 peer reviewers, and the evaluation has been overseen by a Reviewing Editor and Olga Boudker as the Senior Editor. The following individuals involved in review of your submission have agreed to reveal their identity: Andrew J R Plested (Reviewer #1); Grace Brannigan (Reviewer #2).

The reviewers have discussed the reviews with one another and the Reviewing Editor has drafted this decision to help you prepare a revised submission.

Summary:

How structural motions in pLGICs lead to functional channel gating transitions is not well understood. In a previous study (*eLife* 2017), this group used time-resolved fluorescence quenching experiments in a prokaryotic pLGIC, GLIC, and identified motions in the extracellular domain (ECD) that occur prior to channel opening. In this manuscript, the authors extend these studies. To interpret their previous fluorescence quenching data in structural terms, the authors used normal mode analysis to identify structural trajectories underlying GLIC closed to open channel gating transitions. The simulations yielded two pathways leading from closed to open channel end states. To monitor whether motions in the ECD are coupled to motions in the TMD (pore opening), the authors used mutations and the allosteric drug modulator propofol to perturb GLIC gating transitions and compared the effects of these perturbations on proton-induced current responses and steady-state fluorescence quenching data. Based on their data, the authors conclude that they have identified new structural conformations and possible new allosteric pathways during gating, and that GLIC and its mutants have access to a large repertoire of conformational states.

While the data are interesting, the overall concern is with interpretation of the data. The logic of the arguments is not clear enough to support the conclusions. Comparisons between theories are not precise nor quantitative enough to support claim of a new gating model. Simulation description and limitations of this approach are not well described. As written, it is difficult to follow the authors line of reasoning and authors do not discuss alternative mechanisms that can fit their data. Significant re-writing and new analyses to make the arguments clear and focused on what novel contributions the data in this paper are providing are needed. The manuscript would be significantly strengthened by addressing the following major concerns.

Essential revisions:

1) A 1-D, steady state signal (quenching) is used to discern multiple states (meaning, functional conformations, division in space) because it takes multiple values. But the signal is the infinite time average over all states, in different conditions. How do we know that the signal doesn't represent different balances of occupancies of the same small number of states (such as 2 states). There are no apparent reasons why results could not be explained from mutants that change open-closed equilibrium as from new conformational states. The authors need to fully discuss and address this alternative mechanism for explaining the data.

2) What experiments provide direct evidence for intermediates? Can the intermediates states represent abstract transitional milestones, not concrete conformations? That interpretation would be consistent with what has been long established in protein folding, but would entail a major shift in how structures are interpreted, because there would not be a meaningful way to connect conformations to kinetic models.

3) Description of the iMOD fit simulations is unclear and does not provide enough detail. How many simulations were run to yield the two trajectories? It is not clear whether iMod-Fit returns two trajectories from a single calculation or whether multiple simulations were run and the trajectories were clustered. In the methods section (page 25, lines 25-26), trajectory A conformation change is from closed to open whereas trajectory B conformation change is from open to closed. However, in the figures, for both trajectories, frame 1starts at GLIC-pH7 (closed).

4) Since their previous time-resolved fluorescence quenching data (*eLife* 2017) demonstrated that motions monitored at Bimane-136 (ECD β sandwich compaction) and Bim-250 (M2-M3 loop motion) mainly occur prior to channel opening (activation scheme in Figure 4B), it is unclear why the authors conclude that their Bim-250 data support pathway A identified from their simulations. In pathway A, Bim-250-Y197 unquenching happens in same time frame as the increase in pore radius at the 9' position, whereas in pathway B the unquenching of Bim250-Y97 happens before complete pore radius dilation (Figure 2B, 2D). The experimental data at this position seems compatible with the B trajectory. Moreover, the unquenching of Bim-136-W101 appears consistent with either trajectory (Figure 3B) and occurs before the simulated pore dilation. It is unclear what new information the simulations are providing except that the steady-state fluorescence quenching data show good agreement with the simulated end states.

5) To monitor how motions in the ECD are coupled to motions in the TMD (pore opening), the authors used mutations and the allosteric drug modulator propofol to perturb GLIC gating transitions and compared the effects of these perturbations on proton-induced current responses and steady-state fluorescence quenching data. Since the fluorescence data are steady-state, whether a mutation causes an effect on the timing of the structural change in the gating pathway or an effect on the percentages of different conformational states in the ensemble at steady-state is not known and confounds data interpretation. The authors need to discuss this point.

5) Data supporting the statement that mutations of H235, L157 and L246 lead to new global conformations are limited. One could argue that these mutations have dramatic functional effects (eliminate current) and thus, it is not all that surprising that alternative conformations might be adopted that normally would not be visited.

6) Data to support the claim that propofol specifically affects the pre-activation step are limited. Propofol could affect open-closed equilibrium.

7) It is striking that the fluorescence quenching profile for the Q235 mutant with propofol closely mirrors that of H235 without propofol (Black vs light/dashed green lines in Figure 8D). The addition of propofol reverses the effect of the H235Q mutation on structure, not just qualitatively, but close-to-quantitatively, for both the Bim136-Q101W and Bim135-W72 sensors. This is remarkable, especially for the latter sensor where the curve is complex. Yet the discussion treated the two sensors differently despite a similar reversal of the effect of the H235Q mutation, and the authors say the data for Bim135-W72-H235Q cannot be interpreted in structural terms. This explanation is confusing, especially since the corresponding figure is given equal weight in the paper. The authors need to revisit this section to improve clarity or move the mutant results and discussion of explicit technical limitations to supplementary information.

8) The authors claim that their results "challenges the conventional concept that receptor activation involves a single conformational pathway." Do the authors believe their results are inconsistent with the 4 state Heidemann and Changeux models from the 1980s?

9) TMD motions were not measured. Monitoring motion of the M2-M3 loop does not monitor changes in the TMD or pore opening. In previous work, authors used Bimane243 to monitor M2 motions. Additional reporters of TMD motions would be helpful.

[Editors' note: further revisions were suggested prior to acceptance, as described below.]

Thank you for resubmitting your work entitled "Mutational analysis to explore long-range allosteric coupling and decoupling in a pentameric channel receptor" for further consideration by *eLife*. Your revised article has been reviewed by 2 peer reviewers, and the evaluation has been overseen by a Reviewing Editor and Olga Boudker as the Senior Editor. The following individuals involved in review of your submission have agreed to reveal their identity: Andrew J R Plested (Reviewer #1); Grace Brannigan (Reviewer #2).

We appreciate the authors efforts in responding to the previous critiques and in revising the original paper. The revised paper is improved and the reviewers feel that no additional experiments are required. The paper has changed considerably. The biggest concern is that the revised manuscript, as written, is still difficult to follow, even for experts in the field, making it difficult to evaluate and appreciate its impact. The manuscript requires considerable editorial work to improve clarity. The specific points are:

1. The interpretation of the data with propofol is still a weak point. For instance, the manuscript says that "Our data thus shows that propofol does not act locally by altering the conformation of the TMD, but rather acts on the global allosteric transitions by displacing the equilibria and preserving ECD-TMD coupling" However, the preceding discussion is minimal. I spent a while trying to piece together how the data show that propofol's mechanism of action is not local. It is clear that the binding of propofol does affect the conformation far from the TMD, but this sentence implies that the authors have shown that TMD conformational change is insufficient for action. In other words, I could not figure out how the authors showed that ECD conformational shifts were necessary for propofol to act. If I am missing something, a few more sentences spelling out the logic here would be helpful or rewording of conclusions is needed.

2. The new Figure 9 is so full of important information that it gets hard to follow. I request that all the isomerization constants be compiled in a table for ease of comparison. The caption for that table can then refer to the equations and methods from which they were derived and differentiate between normalization schemes. While it may be useful to keep them in Figure 9 as well, Figure 9 is already overwhelmingly complex. I suggest either having fewer plots or fewer elements per plot. The arrows from one plot to the next were also not intuitive. Mainly, I request that the authors look at this figure with fresh eyes and consider breaking it up or streamlining it somehow.

3. Line 37 "Seminal work in the 80s showed that a minimal four-state model describes the main allosteric properties of the muscle-type nAChR (Heidmann and Changeux, 1980; Sakmann et al., 1980)."

First, the word seminal is best avoided; it is rather outdated. I would not take these similarly out-of-date works as a benchmark, rather use them as a counterpoint – perhaps you can say: "although xxx work in the 1980s, …" and then introduce the updates?

4. Line 66 "However, the physiological relevance of these structures or their assignment to particular intermediates or end-states in putative gating pathways remains ambiguous and poorly studied."

This is a very important point and underlines the importance of the work at hand.

5. Line 69 "Conversely, it is likely that key conformations, unfavored by crystal packing lattice or under-represented in receptor populations on cryo-EM grids, are missing in the current structural galleries."

On the other hand, this is overstated. I would say "possible", not likely. Intermediates might be missing, but are they "key"?

6. Line 86 "much faster than ionic current measurement that occurs in the 30-150 millisecond range "

Much faster than the rise time of population or ensemble currents.

7. Line 116 "Two independent trajectories, A and B, were computed starting from each of the two end-state structures and divided into 12 and 11 frames respectively. "

I appreciate that the authors tried to explain better now, but this is still somewhat opaque. Please just say one trajectory goes from rest to active, and the other from active to rest. The use of "end-state structures" is confusing. They are both end and start, depends on which trajectory it is. Or are there really four structures – two crystals and two end states? Some of the figures suggest that each trajectory does not conclude in really the right place. I can't say which one because I have no idea what figure is which (figures not numbered and some do not correspond to figure legend order).

8. With iModFit, I think it is important to discuss how plausible it is that the transitions are not ergodic. This is mentioned in the discussion. In one way, we should not take these trajectories too seriously. But it is also important to consider the possibility that they are pulling out important information from the structures. Are the authors suggesting that the isomerisations are, preferentially, not reversible? I mention below that it would be a great future insight to have non-equilibrium data that could report the non-reversible motion at some of these sites.

9. Later in the paper, the text again makes me feel like I don't understand what iModFit does.

Line 176 "For the ECD quenching pair Bim136-Q101W, the simulations show that Bim136 and the Trp101 indole ring are separated in the resting-like state, and are in close contact in the active-like state (Figure 3A). "

The simulations? Are you referring to the docking results? The resting and active-like states are from structures, aren't they, not from iModfit? If the states used are from structures, there is little predictive power from the docking to these structures that couldn't be deduced by eye, is there? Or are you comparing the state at the end of the iModFit run, which isn't the other crystal state?

Surely iModFit (simulations?) only tells you about the trajectories of the fluorophores? This time-order of transitions between distances is interesting. But why is it mixed up with end state information (surely known from PDB)?

At the very least, a better description is needed. Overall, I still do not understand how the iModFit trajectories help to understand the steady state fluorescence.

10. Line 155 "In conclusion, using iMODfit we could generate two distinct trajectories that are in principle equally plausible to describe a gating transition of GLIC activation. "

But a really key point that doesn't really come up, but I think it should, is that the different trajectories really consist of at least two steps, Twist and the central gating motion, but they occur in different orders. This is a clear appeal to the intermediate states like flip and prime, and motivates the rest of the paper. The role of the compaction is less clear. If there were not distinct movements, the hysteresis in the motions would be much harder to understand. Still, the connection of the iModFit to the steady-state data is less convincing than any non-equilibrium data would be. This is the distinction between a plausible model (as the authors present) and evidence. The change of fluorescence at given sites should have different orders for activation and deactivation, shouldn't it? This would be worth mentioning. It is something for the future of course. And this is not to diminish the insight from the steady-state measurements.

11. Additionally, the flipped state, where the conformational change of the orthosteric site is predicted to be rather complete, but where the channel is closed, would fit the functional requirement of a pre-active state (Lape et al., 2008).

This is trivial because the flip state is just the name of a non-open agonist bound state. Also, flip is not the only type of state that fits, they might all be the same, from different perspectives. I wrote a comment about this once: "Don't flip out: AChRs are primed to catch and hold your attention"

12 The quantitative details of the fitting and the agreement or otherwise seem reasonable but I cannot claim to check in detail, I'm afraid. The individual conformations have various proton bound states and equilibria, so the 3-state model is quite a bit more complicated than at first sight. It might be nice to include the full model (to indicate the assumptions) in a supplementary figure. If, as the authors say, a relatively complicated proton binding scheme is needed to describe even equilibrium data, this is something of a find and needn't be buried. I don't think we have many ideas about how may protons are needed to gate.

13. I did notice that the Y28F mutant has the biggest change in the Pre-open constant. This selective effect was the case for the nearby A52S mutant in the glycine receptor – a big change in flip, no change in the main gating constants (Plested et al., 2007). Quite different at the K276E below that (Lape et al., 2012). But there are tons of mutants on these positions, maybe there are better ones to compare.

14. Table 1 statistics. Multiple mutants are being compared to the same reference value. Unpaired t test is not the correct test to use for these data. Investigators should use an ANOVA with a posthoc test such as a Dunnett or Bonferroni.

15. MWC fitting of the fluorescent and current data is used to conclude that mutations at the ECD alter the pre-activation step while those at the ECD-TMD interface and TMD alter the activation step (gating). Due to the assumption that the mutations do not effect proton affinity to the sites, the authors need to be careful about overinterpreting the data. The modeling provides support but is not conclusive.

16. How the fluorescence quenching data relate to motions identified by iMODfit is not obvious. On page 10, lines 315-318, the authors state "the fluorescence and electrophysiological pH-dependent curves presented in this paper underlie two major allosteric steps, pre-activation (a fast process causing the changes in fluorescence as previously identified in stopped flow experiments and activation (a slower phase). Based on their 2017 *eLife* paper, the bim136 fluorescence reports early pre-gating motions, and bim250 reports early pre-gating motions and some later motions. In the revised manuscript, based on iMOD fit/normal mode analyses, the authors state that bim136 is monitoring a quaternary compaction of the ECD that is occurring throughout the gating cycle and that bim250 is monitoring motion of m2-m3 loop which is part of the 'central gating reorganization' including opening of pore (see page 14, lines 452-455). Later in the paper (page 16, lines 528-529) they state 'ECD compaction is critically involved in pre-activation". This is confusing and requires additional explanation and discussion. If the fluorescent reporters at these positions are monitoring fast, early pre-gating motions then why is the quaternary compaction and m2-m3 loop motion part of the central gating reorganization? Am I missing something?

17. In the abstract, the authors state that 'preactivation involves major asymmetric quaternary motions of the extracellular domain'. It is unclear to me what experimental data support this conclusion. Is this based on the starting pH7.0 crystal structure? The authors need to clarify if the asymmetry that they are describing is at the subunit level or is based on two different motions in the ECD (twisting and compaction). Without strong experimental evidence for asymmetric motions, this conclusion should be removed from the abstract.

18. They use iModFit and NMA as synonyms in some parts of the paper, which causes confusion. iMODfit/Normal mode analysis treats the protein like a 3D elastic network. It doesn't capture interactions with solvent or specific residue-residue interactions. It superimposes multiple local low-energy fluctuations to find likely larger scale conformational fluctuations. You would get asymmetry when it costs less total energy to move a few chains by a lot, than to move all of them by a little. The more chains the protein has, the more likely it is that imodFit will find asymmetry. I'm not sure the imodfit simulations add much regarding asymmetry, but the expected behavior of these macromolecules at room temperature makes it an uncontroversial claim, albeit one without significant new evidence.

19. In revised manuscript (page 5, lines 155-157), authors state 'using iMODfit we could generate two distinct trajectories that are in principle equally plausible to describe a gating transition of GLIC activation'. Additional discussion spelling out the logic here is essential.

It is important for the reader to understand the limitations of iModFit, and for a non-computational reader to know what iModFit is not. The authors need to add further discussion in methods or result sections. It is not a physics-based simulation technique like molecular dynamics – I'd call it a numerical approach for generating hypothetical pathways, and then experiments or simulations need to distinguish between them. They have used it here as a conceptual framework. The software itself is not designed to generate trajectories, but to generate structures. Motions or trajectories generated by Normal Mode Analysis are always reversible. Then imodfit applies a bias on top of that, based on the structure, to get a directional trajectory. They applied two different biases (based on two different structures) so they ended up with two different hypothetical and reversible trajectories.

In their response letter, the authors state "one simulation is starting from the closed conformation to reach the open conformation, and the other from the closed to the open. Both trajectories represent plausible pathways for activation and deactivation." It is unclear whether the authors think that simulation A describes activation (closed channel to open channel) pathway and simulation B describes deactivation pathway (open channel to closed channel) or if they think that both trajectories can describe activation (closed channel to open channel)? Please clarify.

20. The authors should discuss and compare their results from iMODfit/NMA analyses to results from Toby Allen lab (PNAS 2017) using all-atom molecular dynamics with a string method to solve for GLIC gating pathways. What new information has been gained from the iMODfit/NMA?

21. Figures 3 supplementary 1 and 2 and 3 are in in different order compared to figure legends and text on page 6 lines 174-175. Authors need to check the order of the supplementary figures. It would be helpful if figures were labeled for review purposes.

22. Abstract should state which experimental results support their conclusions and describe the novel contributions that the data are providing.

---

## [Author Response]

While the data are interesting, the overall concern is with interpretation of the data. The logic of the arguments is not clear enough to support the conclusions. Comparisons between theories are not precise nor quantitative enough to support claim of a new gating model. Simulation description and limitations of this approach are not well described. As written, it is difficult to follow the authors line of reasoning and authors do not discuss alternative mechanisms that can fit their data. Significant re-writing and new analyses to make the arguments clear and focused on what novel contributions the data in this paper are providing are needed. The manuscript would be significantly strengthened by addressing the following major concerns.

We thank the referees for their evaluation and the insightful general comments. As you will see bellow, we generally agree with most comments, and thus revisited profoundly the interpretation of the data in a more quantitative manner.

To do so, we performed a modeling of the whole set of pH-dependent curves using a 3-states Monod-Wyman-Changeux Model (including resting, pre-active and active states). MWC-type models, calculating the fractional population of the different states in equilibrium conditions, is indeed well suited to simulate the steady state data of fluorescence quenching and electrophysiology (recorded at the current plateau). This modeling procedure is now extensively described at the end of the result section, including two new figures 9 and 10. The resulting set of parameters provides a reasonable and unifying description of most mutant phenotypes. It exemplifies in a quantitative manner the major conclusion that mutations at the ECD principally alter the pre-activation step, while those at the ECD-TMD interface and TMD principally alter the activation step. However, the model also accounts for the phenotype of non-functional mutant (especially H235F), challenging our initial interpretation that those mutants are stabilized in “unique intermediate conformations where motions are decoupled”. Consequently, we do not any more focus the article on “the possibility of multiple conformational pathways during gating”, but rather on the input of our experimental data on the understanding of pre-activation and activation mechanisms. In this context, we further revisited the iModfit data. We agree that the available data do not allow for the discrimination between the two generated trajectories. However, it allows for the assignment of mutant phenotypes to specific protein motions, giving overall a very coherent picture of the key reorganizations involved in pre-activation and activation. Most of the Discussion section was thus rewritten.

Finally, to simplify the flow of the text, we removed from the main text all data related to the Bim135 mutant, since the complexity of the fluorescence data at this level precludes clear conclusions. We however kept this material in the Figure 4 – Supplementary1 to further document effects of mutations and the robustness of the effect of propofol.

Overall, thanks to the referee’s comments, we think we provide now a clearer and much less speculative interpretation of the data. The revised version combines a unique and solid set of fluorescence/electrophysiological data, which are interpreted in the framework of NMA and MWC modeling and which highlight key proteins motions involved in GLIC gating.

Essential revisions:1) A 1-D, steady state signal (quenching) is used to discern multiple states (meaning, functional conformations, division in space) because it takes multiple values. But the signal is the infinite time average over all states, in different conditions. How do we know that the signal doesn't represent different balances of occupancies of the same small number of states (such as 2 states). There are no apparent reasons why results could not be explained from mutants that change open-closed equilibrium as from new conformational states. The authors need to fully discuss and address this alternative mechanism for explaining the data.

In the first version of the article, we interpreted the phenotypes of non-functional mutants proposing that they are stabilized at low pH in new unorthodox conformations. We agree with the general comments of the referee that this idea, while plausible, is one possibility among others, the current data not allowing to draw a firm conclusion. We thus completely removed this idea in the new version.

Following the referee’s advice, we thus tried to fit the entire set of data with a three-state allosteric model, that represents the minimal number of states required to account for the separation of the fluorescence and electrophysiological curves. We show that this model reasonably accounts for the majority of mutants, except for two non-functional ones, L157A and L246A. This may suggest that indeed these mutants adopt unorthodox conformations. However, this idea, still very speculative given the current data, is no more emphasized in the new version.

2) What experiments provide direct evidence for intermediates? Can the intermediates states represent abstract transitional milestones, not concrete conformations? That interpretation would be consistent with what has been long established in protein folding, but would entail a major shift in how structures are interpreted, because there would not be a meaningful way to connect conformations to kinetic models.

The experiments providing direct evidence for intermediate protein “motions” are published in our *eLife* paper of 2017. The two major arguments being the separation of the pH-dependent fluorescence and electrophysiological curves, as well as the kinetic studies, for which a two states model can clearly not account for. The present work further expends this idea, for instance since non-functional mutants still undergo pH-dependent fluorescence changes, another direct demonstration of protein movement not directly related to activation.

The referees ask the question whether such intermediate motion underlie passage through a well-defined single allosteric state, or a multiplicity of conformational pathways. This is an interesting and complex question, that actually is relevant not only to these “intermediate motions”, but also to the end states themselves. In particular, it is more and more admitted that the resting state of GLIC displays high structural flexibility (notably from X-ray, Cryo-EM and MD simulation work cited in the paper). In other word, some of the currently called “allosteric states” actually correspond to a family of conformations. Nonetheless, allosteric models involving discrete states remain useful. While they represent a simplification of the highly complex molecular reorganizations, they are essential to interpret functional data into structural term.

In the new version of the article, we now interpret the data with a 3-state allosteric model. In addition, we discuss that the model does not account for the low cooperativity of the fluorescence curves and we propose, based on NMA analysis, that the implemented intermediate “pre-active” state could correspond to a wider family of states, for instance involving asymmetric movement of individual subunits, in a manner reminiscent of the desensitization of GABAA receptors (Gielen et al., 2020). We also attempt to emphasize in the discussion that the modelling is not meant to demonstrate the occurrence of discrete states, but to refine the interpretation of the mutant phenotypes in term of their relative effect on pre-activation versus activation.

3) Description of the iMOD fit simulations is unclear and does not provide enough detail. How many simulations were run to yield the two trajectories? It is not clear whether iMod-Fit returns two trajectories from a single calculation or whether multiple simulations were run and the trajectories were clustered. In the methods section (page 25, lines 25-26), trajectory A conformation change is from closed to open whereas trajectory B conformation change is from open to closed. However, in the figures, for both trajectories, frame 1starts at GLIC-pH7 (closed).

We thank the reviewer for pointing that the description of the Molecular Modeling methods was not detailed enough, and we tried to fix this issue. The iMOD fit simulations are now detailed in the method section Two simulations were run and the frame numbers have been inverted for one of them in the figures which, we agree, causes confusion. One simulation is starting from the closed conformation to reach the open conformation, and the other from the closed to the open. Both trajectories represent plausible pathways for activation and deactivation. In the result section, we have chosen for clarity to represent both trajectories in the closed to open direction. We propose to clarify and add the missing information as follows:

“Each structure (4NPQ and 4HFI) was fitted, using iMODfit (Lopéz-Blanco and Chacón, 2013), to the simulated electron-microscopy envelope of the other structure. The EM density map resolution was set to 5 Å and the grid size to 0.5 Å. iMODfit was then used to compute two trajectories, fitting each structure to the density of the other structure. In trajectory A structure 4NPQ is fitted to the density of 4HFI, and in trajectory B structure 4HFI is fitted to the density of 4NPQ. The range of modes considered (-n option) was set to 0.5, i.e. half of the modes are considered for the conformational change. The geometry of the ion channel has been computed with MOLEonline webserver (mole.upol.cz), with the ’pore’ mode (Sehnal et al., 2013) We used the FreeRadius value computed at the level of the 9’ residue.”

4) Since their previous time-resolved fluorescence quenching data (eLife 2017) demonstrated that motions monitored at Bimane-136 (ECD β sandwich compaction) and Bim-250 (M2-M3 loop motion) mainly occur prior to channel opening (activation scheme in Figure 4B), it is unclear why the authors conclude that their Bim-250 data support pathway A identified from their simulations. In pathway A, Bim-250-Y197 unquenching happens in same time frame as the increase in pore radius at the 9' position, whereas in pathway B the unquenching of Bim250-Y97 happens before complete pore radius dilation (Figure 2B, 2D). The experimental data at this position seems compatible with the B trajectory. Moreover, the unquenching of Bim-136-W101 appears consistent with either trajectory (Figure 3B) and occurs before the simulated pore dilation. It is unclear what new information the simulations are providing except that the steady-state fluorescence quenching data show good agreement with the simulated end states.

We agree with the referee that discriminating both trajectories on the basis of the current fluorescence data was rather speculative and not clear-cut. We thus removed these considerations in the new version of the paper.

However, iMOD fit simulations still give important information. Regardless of the order in which specific protein motions are appearing in the trajectories, iMOD fit reveals two distinct conformational reorganizations, an asymmetric quaternary reorganization of the ECD, as well as what we call a “central gating motion”, which involves a compaction of the lower part of the β-sandwich, a tilt of M2 toward M3 and in marked increase in the pore radius. Notably, several of the mutations investigated in the paper are predicted to specifically alter some of the key motions involved. iMOD fit simulations thus provides a unique framework to interpret the mutant phenotypes and propose specific protein motions differentially involved in pre-activation versus activation. This analysis is now included in the discussion in the section “Structural interpretation of the mutant phenotypes in the context of NMA trajectories”.

5) To monitor how motions in the ECD are coupled to motions in the TMD (pore opening), the authors used mutations and the allosteric drug modulator propofol to perturb GLIC gating transitions and compared the effects of these perturbations on proton-induced current responses and steady-state fluorescence quenching data. Since the fluorescence data are steady-state, whether a mutation causes an effect on the timing of the structural change in the gating pathway or an effect on the percentages of different conformational states in the ensemble at steady-state is not known and confounds data interpretation. The authors need to discuss this point.

The referees are perfectly correct. In the new version of the manuscript, we interpret the whole series of data using an equilibrium 3-state MWC model, which is adequate to describe the steady state data of fluorescence and electrophysiology (at the plateau of the current). We kept the consideration about the timing of the structural changes at the end of the discussion in a single section (Structural interpretation of the mutant phenotypes in the context of NMA trajectories), that gives overall a coherent and plausible picture of key events involved in gating.

5) Data supporting the statement that mutations of H235, L157 and L246 lead to new global conformations are limited. One could argue that these mutations have dramatic functional effects (eliminate current) and thus, it is not all that surprising that alternative conformations might be adopted that normally would not be visited.

We agree with the referee that data concerning these mutants are limited to conclude that they adopt new conformations. We thus reconsidered their phenotypes, and aimed at integrating them into a single MWC model accounting for all the mutant. Among non-functional mutants, H235F is the best characterized and its X-ray structure shows a well folded conformation, the so-called locally-closed, that is adopted by numerous other GLIC mutants including the WT GLIC (Sauguet et al., 2014). In the course of the fitting procedure, we found actually that we could readily account for the phenotype of this mutant (including shifts in pH_50_s and partial amplitude of the fluorescence curves) by increasing the L_pA_ constant, i.e., by stabilizing the resting state over the pre-active state. Therefore, there is no need to assume a new global conformation in this case. Concerning L157 and L246, as stated in the new version of the article, fits are of lower quality, notably because some of their pH-dependent curves are rather flat and better represented by a straight line. This suggests that those mutants undergo more complex reorganizations than simulated here by a 3-state model, but the data do not allow to conclude on the mechanisms involved. Those two mutants were thus discussed mainly in their ability to preclude the activation transition to occur.

6) Data to support the claim that propofol specifically affects the pre-activation step are limited. Propofol could affect open-closed equilibrium.

To interpret the propofol data, we used the three-state allosteric model, which suggests that propofol affects both the pre-activation and activation equilibria. It is noteworthy that several propofol binding sites have been identified on GLIC by crystallography, and plausibly other unknown sites are present. It is thus difficult to drawn more precise conclusion regarding this matter. In particular our fluorescence sensor reports allosteric effect but would not sense a potential pore blocker effect of propofol suggested in other studies (Fourati et al., 2018).

7) It is striking that the fluorescence quenching profile for the Q235 mutant with propofol closely mirrors that of H235 without propofol (Black vs light/dashed green lines in Figure 8D). The addition of propofol reverses the effect of the H235Q mutation on structure, not just qualitatively, but close-to-quantitatively, for both the Bim136-Q101W and Bim135-W72 sensors. This is remarkable, especially for the latter sensor where the curve is complex. Yet the discussion treated the two sensors differently despite a similar reversal of the effect of the H235Q mutation, and the authors say the data for Bim135-W72-H235Q cannot be interpreted in structural terms. This explanation is confusing, especially since the corresponding figure is given equal weight in the paper. The authors need to revisit this section to improve clarity or move the mutant results and discussion of explicit technical limitations to supplementary information.

In the original version of the manuscript, we have chosen to show data related to the Bim135 sensor to provide a complete description of the quenching data that we have collected. However, as discussed, data related to this position are complex, especially because of the biphasic nature of the fluorescence curve that mix two components. We agree that data related to H235Q is remarkable, but the various mutants tested at this position did not allow us to propose a reasonable interpretation, even a speculative one. Clearly, interesting reorganization are occurring at this level, but they remain mysterious, probably because Bim135 monitor subtle local changes in structure, as illustrated in the docking simulation. To clarify the text, we thus chose to move all the data related to this sensor in the supplementary figures.

8) The authors claim that their results "challenges the conventional concept that receptor activation involves a single conformational pathway." Do the authors believe their results are inconsistent with the 4 state Heidemann and Changeux models from the 1980s?

This claim was removed from the new version. However, not directly linked to this work, a recent paper from the group shows that for the GABAA receptor, asymmetric motions of individual subunit accounts for the biphasic desensitization kinetics (Gielen et al., 2020). It is possible that such a mechanism also applies to nAChRs and other pLGICs and those observations are not necessarily incompatible with a 4 states MWC model.

9) TMD motions were not measured. Monitoring motion of the M2-M3 loop does not monitor changes in the TMD or pore opening. In previous work, authors used Bimane243 to monitor M2 motions. Additional reporters of TMD motions would be helpful.

In the 2017 *eLife* article, we performed mutational work to generate a fluorescent sensor of pore opening. To this aim, based on the Bim243 mutant, we introduced the quenching residue tryptophan at various locations in the upper part of the TMD, nearby the channel gate (figure 3F). Unfortunately, most pairs did not show pH dependent changes in fluorescence, and the only mutant which did so was not functional by electrophysiology. It is clear that mutating this key gating region, especially introducing the bulky tryptophan residue, is detrimental to the gating process. Overall, our work shows that the TriQ technique is powerful to monitor reorganization at the level of solvent-accessible regions of the protein, but not suited to monitor motions in buried locations such as the ion channel.

[Editors' note: further revisions were suggested prior to acceptance, as described below.]

1. The interpretation of the data with propofol is still a weak point. For instance, the manuscript says that "Our data thus shows that propofol does not act locally by altering the conformation of the TMD, but rather acts on the global allosteric transitions by displacing the equilibria and preserving ECD-TMD coupling" However, the preceding discussion is minimal. I spent a while trying to piece together how the data show that propofol's mechanism of action is not local. It is clear that the binding of propofol does affect the conformation far from the TMD, but this sentence implies that the authors have shown that TMD conformational change is insufficient for action. In other words, I could not figure out how the authors showed that ECD conformational shifts were necessary for propofol to act. If I am missing something, a few more sentences spelling out the logic here would be helpful or rewording of conclusions is needed.

The referee is right in pointing out that our conclusion about propofol is somewhat overstated. Indeed, the whole study does not demonstrate that the reorganizations of the ECD are a necessary condition for TMD reorganization and channel opening, although all functional mutants tested do show almost full fluorescence changes at Bim136. We thus reword the conclusion:

“Our data thus shows that propofol does acts on the global allosteric transitions by displacing the equilibria of both pre-activation and activation. It is noteworthy that propofol is also likely to generate local effects upon binding to modulate the function, that are nor investigated here. For instance, its binding into the pore may sterically block ion translocation to produce inhibition”.

2. The new Figure 9 is so full of important information that it gets hard to follow. I request that all the isomerization constants be compiled in a table for ease of comparison. The caption for that table can then refer to the equations and methods from which they were derived and differentiate between normalization schemes. While it may be useful to keep them in Figure 9 as well, Figure 9 is already overwhelmingly complex. I suggest either having fewer plots or fewer elements per plot. The arrows from one plot to the next were also not intuitive. Mainly, I request that the authors look at this figure with fresh eyes and consider breaking it up or streamlining it somehow.

The figure 9 (figure 6 is the new version) has been modified. For clarity the isomerization constants have been summarized in a table and population of the pA state is not shown anymore. To visualize the shifts in isomerization constants, the fits of the reference (Bim136-Q101W or Bim136-Q101W-H235Q in case of Bim136-Q101W-H235Q + propofol) are shown in dashed lines for both current and fluorescence quenching.

3. Line 37 "Seminal work in the 80s showed that a minimal four-state model describes the main allosteric properties of the muscle-type nAChR (Heidmann and Changeux, 1980; Sakmann et al., 1980)."First, the word seminal is best avoided; it is rather outdated. I would not take these similarly out-of-date works as a benchmark, rather use them as a counterpoint – perhaps you can say: "although xxx work in the 1980s, …" and then introduce the updates?

That sentence has been re-phrased “Initially, a minimal 4-states model could describe the main allosteric properties of the muscle-type nAChR”.

4. Line 66 "However, the physiological relevance of these structures or their assignment to particular intermediates or end-states in putative gating pathways remains ambiguous and poorly studied."This is a very important point and underlines the importance of the work at hand.

This point has been repeated in the discussion and abstract.

5. Line 69 "Conversely, it is likely that key conformations, unfavored by crystal packing lattice or under-represented in receptor populations on cryo-EM grids, are missing in the current structural galleries."On the other hand, this is overstated. I would say "possible", not likely. Intermediates might be missing, but are they "key"?

This sentence has been rephrased “Conversely, it is possible that intermediate conformations, unfavored by crystal packing lattice or under-represented in receptor populations on cryoEM grids, are missing in the current structural galleries.”.

6. Line 86 "much faster than ionic current measurement that occurs in the 30-150 millisecond range "Much faster than the rise time of population or ensemble currents.

This sentence has been rephrased “much faster than the rise time of the active population that occurs in the 30-150 millisecond range in electrophysiology recordings”.

7. Line 116 "Two independent trajectories, A and B, were computed starting from each of the two end-state structures and divided into 12 and 11 frames respectively. "I appreciate that the authors tried to explain better now, but this is still somewhat opaque. Please just say one trajectory goes from rest to active, and the other from active to rest. The use of "end-state structures" is confusing. They are both end and start, depends on which trajectory it is. Or are there really four structures – two crystals and two end states? Some of the figures suggest that each trajectory does not conclude in really the right place. I can't say which one because I have no idea what figure is which (figures not numbered and some do not correspond to figure legend order).

To clarify the presentation of the trajectories, we now state explicitly that:

“Two independent trajectories were computed. Trajectory A (12 frames) starts from the closed GLIC-pH7 structure to reach the open GLIC-pH4 structure, and trajectory B (11 frames) starts from the GLIC-pH4 structure to reach the GLIC-pH7 structure. Of note, both trajectories are fully reversible and are equally relevant to describe either activation or deactivation, since normal modes deformation can be applied in the two directions.”

Figures have been re-numbered.

8. With iModFit, I think it is important to discuss how plausible it is that the transitions are not ergodic. This is mentioned in the discussion. In one way, we should not take these trajectories too seriously. But it is also important to consider the possibility that they are pulling out important information from the structures. Are the authors suggesting that the isomerisations are, preferentially, not reversible? I mention below that it would be a great future insight to have non-equilibrium data that could report the non-reversible motion at some of these sites.

We perfectly agree with the referee. This is why we moved the iMODFit to the end of the result section, and presented them more carefully, emphasizing the limitation of the “rough” trajectories and “simple” docking approach. We also state that trajectories, as normal mode deformations, are fully reversible (see answer to point 7). We also agree that these simple computations are pulling out important information from the structure, for instance the “central gating region”, or more locally the reorganization of bimane when attached at position 250.

9. Later in the paper, the text again makes me feel like I don't understand what iModFit does.Line 176 "For the ECD quenching pair Bim136-Q101W, the simulations show that Bim136 and the Trp101 indole ring are separated in the resting-like state, and are in close contact in the active-like state (Figure 3A). "The simulations? Are you referring to the docking results? The resting and active-like states are from structures, aren't they, not from iModfit? If the states used are from structures, there is little predictive power from the docking to these structures that couldn't be deduced by eye, is there? Or are you comparing the state at the end of the iModFit run, which isn't the other crystal state?Surely iModFit (simulations?) only tells you about the trajectories of the fluorophores? This time-order of transitions between distances is interesting. But why is it mixed up with end state information (surely known from PDB)?At the very least, a better description is needed. Overall, I still do not understand how the iModFit trajectories help to understand the steady state fluorescence.

The improper term “simulation” has been removed concerning iMODFit. For this analysis, we considered the extreme intermediates of the trajectories (first and last frames), not the crystallographic structures. This is now stated. Of note, crystallographic structures would show identical results. We do not agree that iMODFit is useless in understanding quenching. For Bim136, the center of mass distances along the trajectories is indeed poorly informative since the bimane moiety, that is exposed to the solvent and rather mobile, adopts multiples poses in the docking procedure generating large fluctuation in center of mass distance. In contrast, the docking procedure is quite interesting for Bim250, which flips from both sides of loop 2, a feature that could not be identified by visual inspection.

10. Line 155 "In conclusion, using iMODfit we could generate two distinct trajectories that are in principle equally plausible to describe a gating transition of GLIC activation. "But a really key point that doesn't really come up, but I think it should, is that the different trajectories really consist of at least two steps, Twist and the central gating motion, but they occur in different orders. This is a clear appeal to the intermediate states like flip and prime, and motivates the rest of the paper. The role of the compaction is less clear. If there were not distinct movements, the hysteresis in the motions would be much harder to understand. Still, the connection of the iModFit to the steady-state data is less convincing than any non-equilibrium data would be. This is the distinction between a plausible model (as the authors present) and evidence. The change of fluorescence at given sites should have different orders for activation and deactivation, shouldn't it? This would be worth mentioning. It is something for the future of course. And this is not to diminish the insight from the steady-state measurements.

It is true that the twist and central gating motions appear in distinctive order in both trajectories. However, we do not have sensors of the twist, so we cannot investigate this global motion in fluorescence experiments. This is why we focused the analysis and discussion on the ECD compaction, M2-M3 movement, and, indirectly from structural considerations, on β-sandwich compaction and M2 movement toward M3. It is also true that the role of the ECD compaction is unclear from iMODFit trajectories. However, the role of the ECD compaction is clear from fluorescence quenching data, that are more solid. This is another reason to present the more speculative iMODFit data at the end of the Results section, and use them only for the interpretation of the fluorescence data. Finally, concerning the hysteresis, both trajectories are theoretically fully reversible, so we think that it is too speculative to discuss the pathways of activation and deactivation in this paper.

11. Additionally, the flipped state, where the conformational change of the orthosteric site is predicted to be rather complete, but where the channel is closed, would fit the functional requirement of a pre-active state (Lape et al., 2008).This is trivial because the flip state is just the name of a non-open agonist bound state. Also, flip is not the only type of state that fits, they might all be the same, from different perspectives. I wrote a comment about this once: "Don't flip out: AChRs are primed to catch and hold your attention"

The mentioned reference has been added.

12. The quantitative details of the fitting and the agreement or otherwise seem reasonable but I cannot claim to check in detail, I'm afraid. The individual conformations have various proton bound states and equilibria, so the 3-state model is quite a bit more complicated than at first sight. It might be nice to include the full model (to indicate the assumptions) in a supplementary figure. If, as the authors say, a relatively complicated proton binding scheme is needed to describe even equilibrium data, this is something of a find and needn't be buried. I don't think we have many ideas about how may protons are needed to gate.

A supplementary figure has been added with a workflow scheme illustrating the steps and assumptions used to build the MWC model. Indeed, the requirement of a second binding site is a new finding that is consistent with our previous study on GLIC proton site where we could not find a single proton site that would abolish activation when mutated (Nemecz et al., 2017).

13. I did notice that the Y28F mutant has the biggest change in the Pre-open constant. This selective effect was the case for the nearby A52S mutant in the glycine receptor – a big change in flip, no change in the main gating constants (Plested et al., 2007). Quite different at the K276E below that (Lape et al., 2012). But there are tons of mutants on these positions, maybe there are better ones to compare.

In the examples of pathological mutations used in the discussion we focused on the mutation homologous to the ones used on GLIC. The papers mentioned here are studying other residues but their conclusions are in line with the residues we studied in the same regions. We added them at the end of the discussion:

“Interestingly, on the glycine receptor α1, other mutations have been studied by single channel recordings and are described to affect principally a flip pre-activation-like step for A52S in the loop 2 at the ECD-ECD interface (Plested et al., 2007) or gating for K276E on the M2-M3 loop (Lape et al., 2012). These data suggest that mutations produce similar allosteric perturbations on GLIC and GlyR in those regions.”

14. Table 1 statistics. Multiple mutants are being compared to the same reference value. Unpaired t test is not the correct test to use for these data. Investigators should use an ANOVA with a posthoc test such as a Dunnett or Bonferroni.

A one-way ANOVA followed by a Dunnett test has been performed on data where mutants were compared to a common reference (Bim136-Q101 of Bim250-Y197). To compare individually fluorescence to electrophysiological pH_50_ for each mutant, we kept the t tests.

15. MWC fitting of the fluorescent and current data is used to conclude that mutations at the ECD alter the pre-activation step while those at the ECD-TMD interface and TMD alter the activation step (gating). Due to the assumption that the mutations do not effect proton affinity to the sites, the authors need to be careful about overinterpreting the data. The modeling provides support but is not conclusive.

Indeed, choosing to modify isomerization constants and not affinities is a choice made in this model and conclusions were already mitigated in the discussion:

“We also postulated that the various mutants only alter the isomerization constants between states. However, the dataset does not allow for the discrimination between effect on binding affinity versus isomerization constants. The effects of mutations on the isomerization constants are thus meant here to evaluate the global effect of the mutations on pre-activation versus activation, but it is possible that they actually underlie alteration of isomerization constant, affinity constants, or both.”

We now develop further the idea:

“Among the various mutations investigated here, E26Q, E222Q, H235F/Q neutralize the charge of titratable amino acids. It is thus possible that in these cases the mutation eliminates a proton binding site. However, a local impact of a mutation on a proton binding site, or on a set of inter-residues interactions altering the allosteric equilibria, will be equally valid in assigning local structural alterations to pre-active/active phenotypes.”

16. How the fluorescence quenching data relate to motions identified by iMODfit is not obvious. On page 10, lines 315-318, the authors state "the fluorescence and electrophysiological pH-dependent curves presented in this paper underlie two major allosteric steps, pre-activation (a fast process causing the changes in fluorescence as previously identified in stopped flow experiments and activation (a slower phase). Based on their 2017 eLife paper, the bim136 fluorescence reports early pre-gating motions, and bim250 reports early pre-gating motions and some later motions. In the revised manuscript, based on iMOD fit/normal mode analyses, the authors state that bim136 is monitoring a quaternary compaction of the ECD that is occurring throughout the gating cycle and that bim250 is monitoring motion of m2-m3 loop which is part of the 'central gating reorganization' including opening of pore (see page 14, lines 452-455). Later in the paper (page 16, lines 528-529) they state 'ECD compaction is critically involved in pre-activation". This is confusing and requires additional explanation and discussion. If the fluorescent reporters at these positions are monitoring fast, early pre-gating motions then why is the quaternary compaction and m2-m3 loop motion part of the central gating reorganization? Am I missing something?

In our *eLife* 2017 paper, we showed for Bim136-Q101W and Bim250-Y197 that the majority of the changes in fluorescence occur with very fast kinetics. Bim250 indeed shows some fluorescence changes appearing with slower kinetics, but they contribute by less than 20% to the total ∆F, and thus were neglected in the present steady-state study. Still, Bim250 monitors a pre-activation motion by fluorescence, but the modifications themselves alter principally the activation transition (stronger effect on the pH_50_ of ∆I as compared to ∆F). We now emphasize this point in the discussion by stating:

“The mutational analysis also shows for most mutations mixed effects on the isomerization constants of activation and pre-activation, suggesting that both processes involve overlapping regions. The Bim250 position is noteworthy in this respect, since the bimane, reporting an outward motion of the M2-M3 loop, monitors pre-activation, while the modification itself (P250C mutation plus reaction with bimane) principally alters the activation process. It is thus plausible that the M2-M3 loop could move in two successive steps, a first one during pre-activation conditioning dequenching and a second one during activation. In either case, our data further highlight a central role for this loop in ECD-TMD coupling.”

We then later in the text speculate about one possible succession of reorganizations:

“Concerning the order in which reorganizations are observed, Trajectory A fits much better the fluorescence data. It suggests a scenario involving, during pre-activation, progressive ECD compaction and beginning of the M2-M3 loop motion, generating the fluorescence variations. Then the M2-M3 loop completes its movement in concert with β-sandwich compaction and pore opening. Future computational studies are needed to investigate this possibility.”

17. In the abstract, the authors state that 'preactivation involves major asymmetric quaternary motions of the extracellular domain'. It is unclear to me what experimental data support this conclusion. Is this based on the starting pH7.0 crystal structure? The authors need to clarify if the asymmetry that they are describing is at the subunit level or is based on two different motions in the ECD (twisting and compaction). Without strong experimental evidence for asymmetric motions, this conclusion should be removed from the abstract.

The abstract has been re-written leaving out the notion of asymmetry, which we agree is not addressed in the fluorescence experiments.

18. They use iModFit and NMA as synonyms in some parts of the paper, which causes confusion. iMODfit/Normal mode analysis treats the protein like a 3D elastic network. It doesn't capture interactions with solvent or specific residue-residue interactions. It superimposes multiple local low-energy fluctuations to find likely larger scale conformational fluctuations. You would get asymmetry when it costs less total energy to move a few chains by a lot, than to move all of them by a little. The more chains the protein has, the more likely it is that imodFit will find asymmetry. I'm not sure the imodfit simulations add much regarding asymmetry, but the expected behavior of these macromolecules at room temperature makes it an uncontroversial claim, albeit one without significant new evidence.

The asymmetric nature of the transition has been removed from the abstract and just discussed at the end of the discussion in the section “Speculative interpretation of the mutant phenotypes in the context of computational trajectories”*.*

19. In revised manuscript (page 5, lines 155-157), authors state 'using iMODfit we could generate two distinct trajectories that are in principle equally plausible to describe a gating transition of GLIC activation'. Additional discussion spelling out the logic here is essential.It is important for the reader to understand the limitations of iModFit, and for a non-computational reader to know what iModFit is not. The authors need to add further discussion in methods or result sections. It is not a physics-based simulation technique like molecular dynamics – I'd call it a numerical approach for generating hypothetical pathways, and then experiments or simulations need to distinguish between them. They have used it here as a conceptual framework. The software itself is not designed to generate trajectories, but to generate structures. Motions or trajectories generated by Normal Mode Analysis are always reversible. Then imodfit applies a bias on top of that, based on the structure, to get a directional trajectory. They applied two different biases (based on two different structures) so they ended up with two different hypothetical and reversible trajectories.In their response letter, the authors state "one simulation is starting from the closed conformation to reach the open conformation, and the other from the closed to the open. Both trajectories represent plausible pathways for activation and deactivation." It is unclear whether the authors think that simulation A describes activation (closed channel to open channel) pathway and simulation B describes deactivation pathway (open channel to closed channel) or if they think that both trajectories can describe activation (closed channel to open channel)? Please clarify.

The iMODfit procedure, and its limitations, have been presented in detail in the results and method section to clarify the various concerns raised here and in previous questions.

20. The authors should discuss and compare their results from iMODfit/NMA analyses to results from Toby Allen lab (PNAS 2017) using all-atom molecular dynamics with a string method to solve for GLIC gating pathways. What new information has been gained from the iMODfit/NMA?

The discussion now includes an overview of the gating trajectories generated *in silico*, in the section “Speculative interpretation of the mutant phenotypes in the context of computational trajectories”. We started with all-atom MDs, notably the work from Allen and co-workers, and then move to iMODfit trajectories explicitly stating that:

“While these trajectories are rough and do not implement fine atomistic interactions, they allow to visualize plausible collective motions in relation with the reorganization of the quenching pairs”.

21. Figures 3 supplementary 1 and 2 and 3 are in in different order compared to figure legends and text on page 6 lines 174-175. Authors need to check the order of the supplementary figures. It would be helpful if figures were labeled for review purposes.

Figures have been re-labeled.

22. Abstract should state which experimental results support their conclusions and describe the novel contributions that the data are providing.

The abstract has been re-written.